# Designing multi-metal-site nanosheet catalysts for $CO_2$ photoreduction to ethylene

Xiaodong Li [1,7], Li Li [2,7], Xiaohui Liu[3,7], Jiaqi Xu [4,5,7], Xingyuan Chu[3], Guangbo Chen [3], Dongqi Li [3], Mingchao Wang [3], Xia Wang [6], Chandrasekhar Naisa[1], Jing Gao [4], Yongfu Sun [2] ✉, Michael Grätzel [4] ✉ & Xinliang Feng [1,3] ✉

Catalysts featuring multiple active sites hold significant potential for $CO_2$ photoconversion to multi-carbon products. However, multi-metal-site catalysts typically face challenges with low yields and selectivity for ethylene production, with a lack of definitive design guidelines. Here we show that Bader charge can serve as a critical descriptor for delineating the structure–activity relationship of kesterite-like nanosheets in the reduction of $CO_2$ to ethylene. We propose the Bader-Regulate-Performance principle – apposite Bader charge can provide a moderate energy barrier for intermediate adsorption and C-C coupling simultaneously, thus promoting the performance for ethylene generation. Among the predicted multi-metal-site nanosheets, the $Cu_2ZnSnS_4$, with the appropriate Bader charge, achieves a high ethylene yield of 25.16 $\mu mol\, g^{-1}\, h^{-1}$ with electron selectivity of 72.4% under visible light irradiation, surpassing those of reported photocatalysts under similar catalytic conditions. Our findings provide crucial insights into the design of efficient catalysts for photocatalytic $CO_2$ conversion to multi-carbon products.

Photocatalytic $CO_2$ conversion provides a promising approach to tackling the increasing carbon emissions, greenhouse effects, and energy crisis[1]. Among numerous products, ethylene ($C_2H_4$) stands out due to its relatively high market value, demonstrating unique advantages compared to single-carbon ($C_1$) products[2]. Currently, ~75% of petrochemical products originate from $C_2H_4$[3]. The global demand for $C_2H_4$ is estimated to surpass 150 million tons annually, with its market value exceeding 100 billion US dollars in 2020[4,5]. The prevalent industrial process for $C_2H_4$ production involves the high-temperature steam cracking of naphtha in the temperature range of 800 °C–900 °C, resulting in substantial energy consumption and environmental repercussions[6]. Regrettably, only a few photocatalysts,

such as $AuPd/TiO_2$ and $FeCoS_2$, have shown the potential for $CO_2$ conversion into $C_2H_4$ under mild conditions[7]. However, these photocatalysts still grapple with challenges such as low formation rates and suboptimal selectivity for $C_2H_4$ (yield <20 $\mu mol\, g^{-1}\, h^{-1}$, selectivity <60%). Therefore, developing efficient catalysts for the selective photoreduction of $CO_2$ to $C_2H_4$ under mild conditions is imperative, aligning with the pursuit of sustainable and environmentally friendly processes.

Two-dimensional (2D) transition-metal compounds demonstrate notable photocatalytic $CO_2$ activity, owing to their abundant surface-active sites and efficient charge carrier separation[8,9]. Concurrently, the multi-metal sites on these catalysts exert precise control over the

[1]Max Planck Institute of Microstructure Physics, Weinberg 2, Halle, Germany. [2]Hefei National Research Center for Physical Sciences at Microscale, University of Science and Technology of China, Hefei, P. R. China. [3]Faculty of Chemistry and Food Chemistry & Center for Advancing Electronics Dresden (cfaed), Dresden University of Technology, Dresden, Germany. [4]Laboratory of Photonics and Interfaces, École Polytechnique Fédérale de Lausanne, Lausanne, Switzerland. [5]Key Laboratory of Green Chemistry & Technology of Ministry of Education, College of Chemistry, Sichuan University, Chengdu, Sichuan, P. R. China. [6]Max Planck Institute for Chemical Physics of Solids, Dresden, Germany. [7]These authors contributed equally: Xiaodong Li, Li Li, Xiaohui Liu, Jiaqi Xu. ✉e-mail: yfsun@ustc.edu.cn; michael.graetzel@epfl.ch; Xinliang.Feng@tu-dresden.de

electronic structure of the active site and the adsorption configuration of intermediates during $CO_2$ reduction, therefore regulating the selectivity to different types of bonding products[10,11]. For example, Chen et al. fabricated a 2D bimetallic oxyhalide $Pb_{0.6}Bi_{1.4}O_2Cl_{1.4}$ and achieved a performance for $CO_2$ photoreduction to methanol[12]. They also utilized the Ag-Cu Lewis acid-base dual sites on the surface of $Ag_2Cu_2O_3$ nanowires to realize photoconversion of $CO_2$ to methane[13]. Besides, multi-metal sites can also break the linear-scaling relationship and promote the generation of multi-carbon ($C_{2+}$) products, like the designed Ni-V sites for ethane production[14–16]. Hence, 2D transition-metal catalysts, featuring multi-metal sites, are highly promising candidates for advancing the field of photocatalytic $CO_2$ conversion into $C_2H_4$. Nevertheless, there is currently a lack of clear design guidelines for designing multi-metal-site catalysts. It is known that CO* adsorption and C-C coupling processes are crucial for ethylene production in $CO_2$ reduction[17]. However, there are two key issues that need to be addressed: (1) how does CO* adsorption specifically relate to C-C coupling? Does stronger CO* adsorption result in a lower energy barrier for C-C coupling? (2) How to simultaneously regulate CO* adsorption energy and C-C coupling energy barriers? There is still no descriptor to explain the relationship between material properties and these two important processes. Therefore, a suitable descriptor for predicting the relationship with CO* adsorption and C-C coupling processes would significantly facilitate the catalyst development in producing ethylene.

Herein, we discover that the Bader charge can serve as a descriptor to predict the performance of kesterite-like $Cu_2M_iM_jS_4$ ($M_i$ = Mn, Fe, Co, Ni, Zn; $M_j$ = Ge, Sn) catalysts for $CO_2$ reduction into ethylene. Bader charge determines the number of electrons associated with each atom by partitioning the electron density between atoms, which normally reflects electronic structure of the material and its interaction with surrounding atoms[18]. By analyzing the electronic structure, adsorption energy, and translation state (TS) energy barrier of C-C coupling processes, we find that both the energy for CO* adsorption and C-C coupling exhibits a linear relationship with the Bader charge of the Cu atom, but in the opposite trend. In this respect, the smallest Bader charge loss of Cu atoms results in the strongest CO* adsorption energy but also leads to the highest C-C coupling energy barrier, which contrasts with previous reports suggesting that strong CO* adsorption energy favors the C-C coupling process[17]. We thus propose the Bader-regulate-performance (BRP) principle, that is, a moderate Bader charge can effectively balance CO* adsorption and C-C coupling processes, optimizing them to suitable levels (neither too strong nor too weak), thereby facilitating $CO_2$ reduction to produce $C_{2+}$ products, akin to the Sabatier principle[19]. We further confirm that $M_i$ and $M_j$ atoms along with sulfur vacancies ($S_v$) can coordinately regulate the Bader charge of Cu atoms. We verify the theoretical predictions through experiments by synthesizing various 2D kesterite-like nanosheets, such as $Cu_2MnSnS_4$ nanosheet (CMTS), $Cu_2CoSnS_4$ nanosheet (CCTS), $Cu_2NiSnS_4$ nanosheet (CNTS), and $Cu_2ZnSnS_4$ nanosheet (CZTS). Among them, the CZTS nanosheet, predicted to have a suitable Bader charge (−0.42) in Cu atoms, achieves a $C_2H_4$ generation rate of 25.16 μmol $g^{-1}$ $h^{-1}$ and an electron selectivity of 72.4% under visible-light irradiation, which is superior to the previous reports (<20 μmol $g^{-1}$ $h^{-1}$) under similar catalytic conditions.

## Results

### Theoretical design of multi-site $Cu_2M_iM_jS_4$ nanosheets

To explore the kesterite-like $Cu_2M_iM_jS_4$ nanosheets theoretically, we built stable 2D slab models according to the surface formation energy (Supplementary Fig. 1-4). Considering that anion vacancies often lead to local charge accumulation, which can stabilize the reaction intermediates, thereby enhancing catalytic activity and regulating product selectivity[20], we further calculated the formation energy of surface $S_v$. We found that the $S_v$ can be spontaneously generated on the surface of

2D $Cu_2M_iM_jS_4$ slabs (except the $Cu_2FeSnS_4$ (CFTS)) with a negative $S_v$ formation energy. We then calculated the Bader charge of the pristine and $Cu_2M_iM_jS_4$ slabs with sulfur vacancy ($Cu_2M_iM_jS_4$-$S_v$), as shown in Supplementary Fig. 5-12. The formation energy of $S_v$ and Bader charge of Cu atoms in the pristine $Cu_2M_iM_jS_4$ slabs show an inverted volcano relationship (Fig. 1a), in which the slab with Cu atoms possessing Bader charge of −0.528 (nearby the CNTS and CZTS) is more likely to form $S_v$ theoretically. In the pursuit of $C_{2+}$ products, the processes of CO* adsorption and C-C coupling are pivotal[21,22]. We then calculated the adsorption energy of CO* intermediates ($E_{ad}$(CO*)) at various potential sites within the CZTS-$S_v$ slab. As depicted in Fig. 1b and Supplementary Figs. 13-14, the model in which CO is adsorbed at site 3 (Cu site near the $S_v$) exhibits the lowest adsorption energy of −0.43 eV. This negative and strong adsorption energy signifies that CO* intermediates can spontaneously bond with the Cu sites in the CZTS-$S_v$ slab, implying an extended residence time on the surface[23].

Normally, the stronger adsorption of CO* species, in turn, provides a greater likelihood for C-C coupling. We then thoroughly analyzed the C-C coupling processes (Supplementary Figs. 15–17), including the three typical pathways: CO-CO, CO-CHO, and CO-COH. Among them, the CO-COH route with the lowest coupling energy barrier was suggested as the most likely C-C coupling pathway in our case. However, after calculating the climbing image-nudged elastic band (CI-NEB) TS energy barrier during the C-C coupling processes of CO* and COH* ($E_{C-C}$) across the $Cu_2M_iM_jS_4$-$S_v$ slabs (Supplementary Figs. 18–21), we found that both the $E_{ad}$(CO*) and $E_{C-C}$ exhibit a linear relationship with the Bader charge of the Cu atoms near the $S_v$, while they show the totally opposite trend. This means that a smaller Bader charge could lead to a stronger CO* adsorption, but at the same time make C-C coupling more difficult. In contrast, a moderate Bader charge could effectively balance CO* adsorption and C-C coupling processes and optimize them to suitable levels (neither too strong nor too weak), facilitating $CO_2$ reduction to produce $C_{2+}$ products; we call this the BRP principle. Therefore, the Bader charge of Cu atoms in the $Cu_2M_iM_jS_4$-$S_v$ slabs can serve as a descriptor to predict their performance for $CO_2$ reduction to $C_{2+}$ products. The ideal photocatalysts with Bader charge around 0.42 on unsaturated Cu sites in the orange circle area of Fig. 1c, like CZTS slabs, could provide moderate $E_{ad}$(CO*) and $E_{C-C}$ simultaneously, possessing a higher potential for $C_{2+}$ generation from $CO_2$ reduction according to the BRP principle. To explore the key factors regulating the Bader charge of Cu atoms in the $Cu_2M_iM_jS_4$-$S_v$ slabs, we constructed analogous theoretical models for CuS, $CuZnS_2$, and $Cu_3SnS_4$ slabs. A distinct linear relationship is still evident, as depicted in Supplementary Fig. 22-23, where the $E_{ad}$(CO*) increases incrementally with the reduction in Bader charge loss [CZTS (|−0.42|) < $Cu_3SnS_4$ (|−0.47|) < CuS (|−0.52|) < $CuZnS_2$ (|−0.54|)]. The above theoretical results imply that $M_i$ and $M_j$ atoms, along with $S_v$, can collaboratively modulate the Bader charge of Cu atoms.

To analyze the suitability of $Cu_2M_iM_jS_4$-$S_v$ nanosheets for $CO_2$ photoreduction, we took the CZTS slab as an example to study the band structure. As displayed in Supplementary Fig. 24, the conduction band maximum (CBM) of the CZTS-$S_v$ slab surpasses the potential required for $CO_2$ reduction (e.g., 0.14 V for $C_2H_4$ generation), while the valence band minimum (VBM) falls below the potential for $H_2O$ oxidation (e.g., 1.23 V for $O_2$ generation). This result suggests the capability of CZTS-$S_v$ slab for simultaneous $CO_2$ reduction and $H_2O$ oxidation. As a proof of concept, we evaluated the performance of CZTS-$S_v$ slab for $CO_2$ reduction to $C_2H_4$ in theory. The comprehensive calculations of Gibbs free energy for $CO_2$ reduction to $C_2H_4$ were conducted to unveil the rate-determining step (RDS) and dynamic conversion processes. As depicted in Fig. 1d, the elementary step of $CO_2 \rightarrow COOH*$ is identified as the RDS with Gibbs free energy of 1.52 eV, while the energy barrier for C-C coupling (CO* + COH* → CO-COH*) is as low as 0.91 eV. That means the thermodynamic energy barrier for $C_2H_4$ production is induced by the protonation of $CO_2$ to COOH*, and the

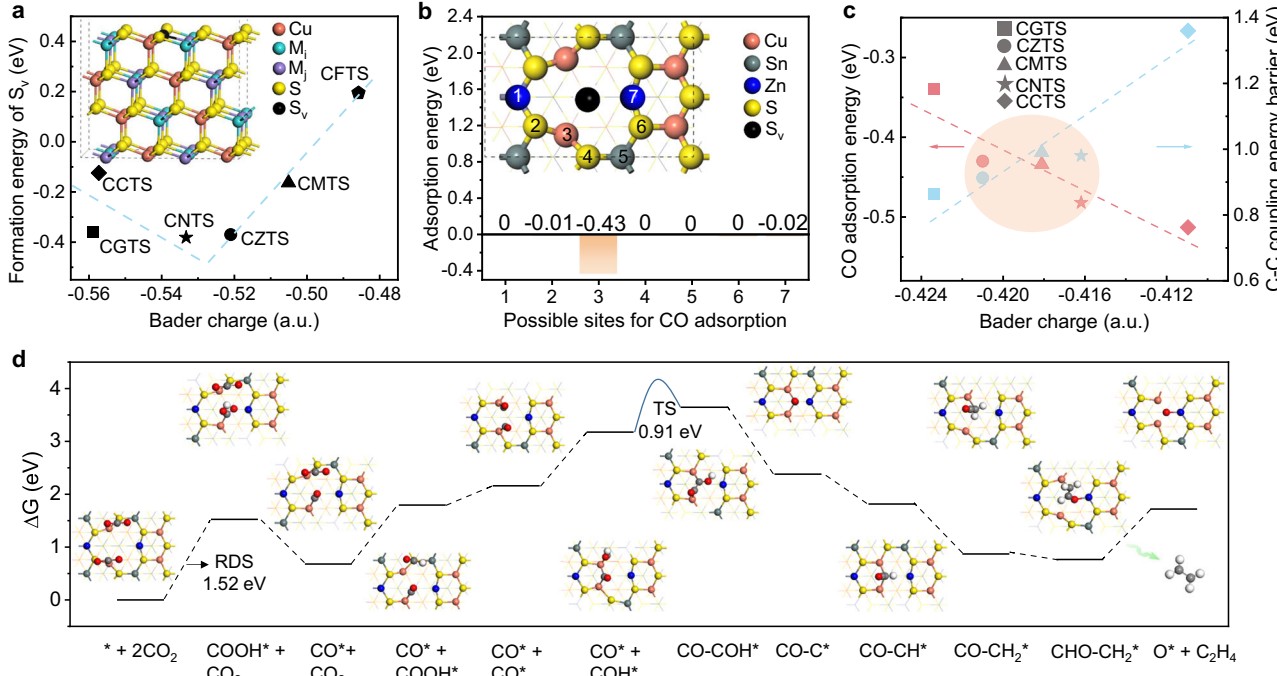

**Fig. 1 | Theoretical prediction of the multi-site Cu₂MᵢMⱼS₄ nanosheets. a** Plots of formation energy of $S_v$ changing with Bader charge of Cu atom in pristine Cu₂MᵢMⱼS₄ slab, in which the formation energy of $S_v$ and Bader charge of Cu atoms in the pristine Cu₂MᵢMⱼS₄ slabs show an inverted volcano relationship. **b** $E_{ad}(CO^*)$ at different sites of the CZTS-$S_v$ slab. **c** CO* adsorption energy ($E_{ad}(CO^*)$) (red line) and C-C coupling energy barrier ($E_{C-C}$) (blue line) changing with the Bader charge of Cu atom in Cu₂MᵢMⱼS₄-$S_v$ slab, in which both the $E_{ad}(CO^*)$ and $E_{C-C}$ exhibit a linear relationship with the Bader charge of the Cu atoms near the $S_v$ but in opposite trend. **d** Gibbs free energy of each step during the $CO_2$ reduction to ethylene over the CZTS-$S_v$ slab, in which the energy barrier of the rate-determining step (RDS) is 1.52 eV. Source data are provided as a Source Data file.

C-C coupling process is not the determining factor, while it is relatively easier to proceed on the CZTS-$S_v$ slab.

### Synthesis and characterization of Cu₂MᵢMⱼS₄-$S_v$ nanosheets

Inspired by the above theoretical predictions, we then synthesized a series of Cu₂MᵢMⱼS₄-$S_v$ (Mᵢ = Mn, Co, Ni, Zn; Mⱼ = Sn) nanosheets. As shown in Fig. 2a–d, transmission electron microscopy (TEM) images show that the Cu₂MᵢMⱼS₄-$S_v$ nanosheets possess flake-like morphology. High-resolution TEM (HRTEM) images verify that CMTS-$S_v$, CCTS-$S_v$, and CZTS-$S_v$ nanosheets possess an exposed surface along (112) direction, consistent with the predicted facet with the lowest surface formation energy (112-S) (Supplementary Fig. 25), while CNTS-$S_v$ nanosheet shows the (111) surface due to its cubic crystallinity. Energy dispersive spectroscopy (EDS) element mapping images illustrate the homogenous distribution of Cu, Mᵢ, Mⱼ, and S elements. Figure 2e further confirms the nanosheet configuration of CNTS-$S_v$ with a thickness of -8.57 nm (around 7 unit cells). XRD patterns in Fig. 2f and Supplementary Fig. 26 depict that the phase of CMTS-$S_v$, CCTS-$S_v$, CNTS-$S_v$, and CZTS-$S_v$ nanosheets is in line with the PDF number of 51-0757, 26-0513, 26-0552, and 26-0575, respectively. In contrast, bulk CZTS refers to the standard Cu₂ZnSnS₄ compound that is also synthesized (Supplementary Fig. 27).

To confirm the existence of $S_v$ within the Cu₂MᵢMⱼS₄-$S_v$ nanosheets, we carried out the electron spin resonance (ESR) spectrum, as shown in Fig. 2g and Supplementary Fig. 28. The ESR signal at $g = 2.003$ can be assigned to the presence of $S_v$[24]. The presence of $S_v$ in the CZTS nanosheet is further validated by a 0.25 eV upshift of the S 2$p$ peak in the X-ray photoelectron spectra (XPS) relative to that of bulk CZTS (Fig. 2h), while the Cu 2$p$, Zn 2$p$ and Sn 3$d$ XPS peaks in Supplementary Fig. 29-30 remain almost unchanged[25]. Besides, the Fourier-transform infrared (FTIR) spectrum (Supplementary Fig. 31a) illustrates that the CZTS nanosheet exhibits the same characteristic peaks as the bulk, with no obvious peaks related to organic compounds, which indicates

the successful synthesis of clean 2D CZTS nanosheet[26]. However, the CZTS nanosheet lacks distinct Raman shifts at 289, 339, and 362 cm⁻² that are present in the bulk (Supplementary Fig. 31b)[27]. This absence suggests a poor crystallinity of the nanosheet sample due to abundant S defects[28]. EDS analysis of element content in Supplementary Fig. 32 directly demonstrates a reduced S proportion in the CZTS nanosheet, in which atomic percent (at.%) of S element is 34.8% in the CZTS nanosheet while 45.0% in the bulk. Furthermore, X-ray absorption near-edge structure (XANES) analyses were performed to determine the valence and coordination environment. As depicted in Fig. 2i and the insert, compared with bulk CZTS, the Cu K-edge position of the CZTS nanosheet slightly shifts to a lower energy, which verifies a lower coordination of Cu atoms in CZTS nanosheet, further confirming the presence of the $S_v$[29]. Additionally, we analyzed the synthesis cost of CZTS nanosheet and found that the sample could be easily synthesized to more than 2 grams at a low cost using the solvothermal method (Supplementary Fig. 31c), demonstrating its scalability and cost-effectiveness (Supplementary Note 4).

### Photocatalytic $CO_2$ conversion by Cu₂MᵢMⱼS₄-$S_v$ nanosheets

The good light absorption capacity and suitable band edge position of the catalyst are the prerequisites for photocatalytic $CO_2$ reduction[30]. UV-vis-NIR diffuse reflectance spectra in Fig. 3a show that the CZTS-$S_v$ nanosheet presents a much better light absorption ability than the bulk counterpart. The corresponding Tauc plots (insert image in Fig. 3a) suggest the band gaps of 1.81 eV and 1.48 eV for the CZTS-$S_v$ nanosheet and bulk, respectively. Supplementary Fig. 33 also illustrates that CMTS-$S_v$, CCTS-$S_v$, and CNTS-$S_v$ nanosheets possess a suitable band gap (CMTS-$S_v$: 1.60 eV; CCTS-$S_v$: 1.84 eV; CNTS-$S_v$: 1.71 eV) for $CO_2$ reduction with a good absorption ability for visible light. Combined with the ultraviolet photoelectron spectroscopy (UPS) (Fig. 3b and Supplementary Fig. 34), we identify the band structure of the CZTS-$S_v$ nanosheet. As shown in Fig. 3c, the experimental CBM

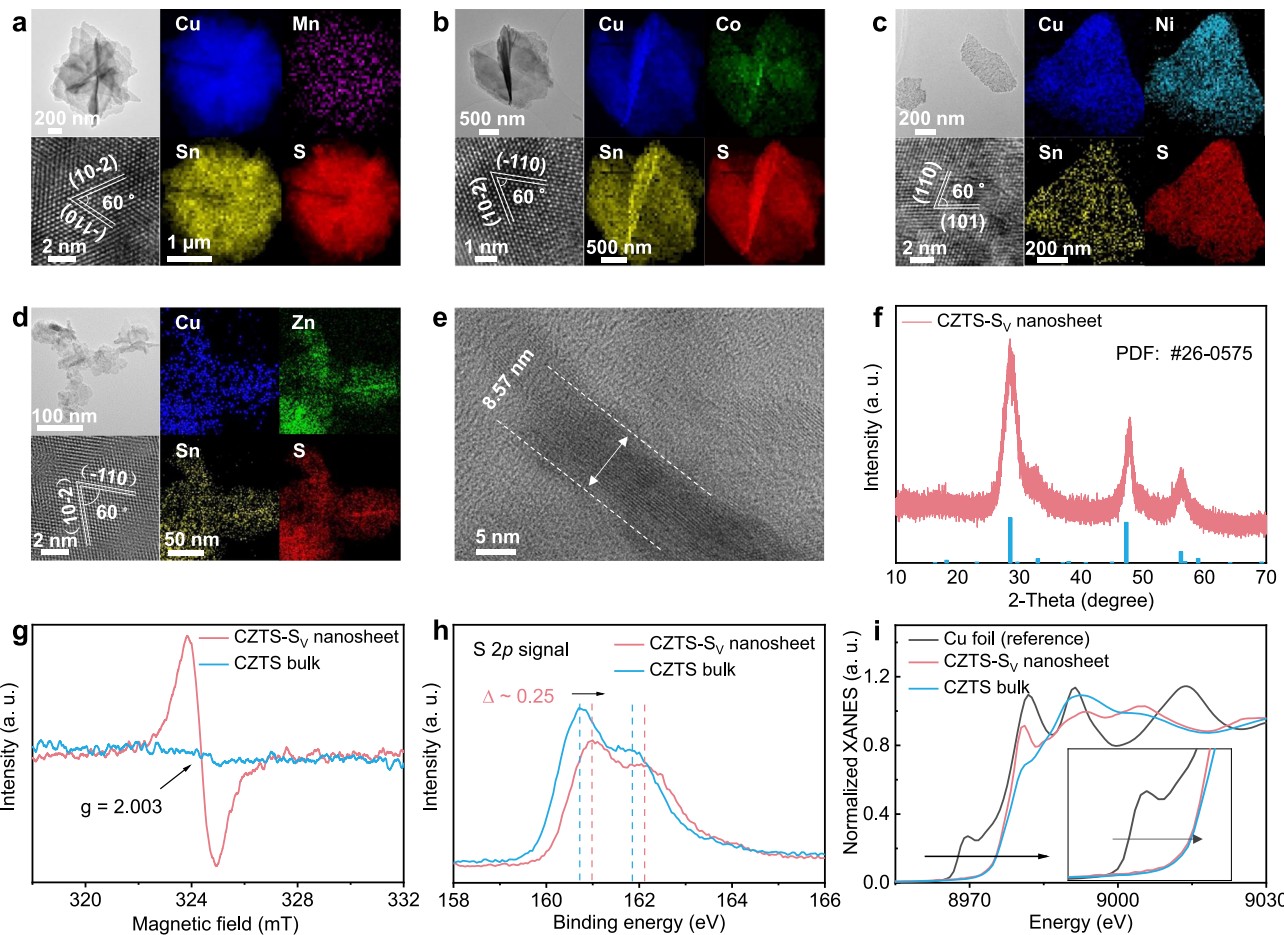

**Fig. 2 | The experimental characterizations of Cu₂MᵢMⱼS₄-Sᵥ nanosheets.** TEM, HRTEM and EDS mapping images for **a** CMTS-Sᵥ nanosheet, **b** CCTS-Sᵥ nanosheet, **c** CNTS-Sᵥ nanosheet and **d** CZTS-Sᵥ nanosheet, in which the exposed facet can be inferred along [112] direction ([111] direction for CNTS-Sᵥ nanosheet) because interplanar distances match well with the $d_{110}$ and $d_{10\text{-}2}$ spacings ($d_{110}$ and $d_{101}$ spacings for CNTS-Sᵥ nanosheet), and the corresponding dihedral angle of 60° agrees well with the calculated angle between the (−110) and (10-2) planes ((110) and (101) planes for CNTS-Sᵥ nanosheet). **e** TEM image for CZTS-Sᵥ nanosheet, in which the thickness of the obtained nanosheet is determined to be 8.57 nm according to the side view of the sample. **f** XRD pattern of CZTS-Sᵥ nanosheet. **g** Electron spin resonance (ESR) spectra of the CZTS bulk and nanosheet, in which the signal at $g = 2.003$ corresponds to the Sᵥ. **h** XPS spectra of S 2$p$ for the CZTS bulk and nanosheet. **i** Cu K-edge XANES spectra of the CZTS bulk and nanosheet. Source data are provided as a Source Data file.

(−0.19 V) and VBM (1.62 V) are well consistent with the above theoretical values (−0.16 V for CBM and 1.64 V for VBM), which confirm the ability of CZTS-Sᵥ nanosheet for CO₂ reduction and H₂O oxidation simultaneously.

To evaluate the performance of the obtained Cu₂MᵢMⱼS₄-Sᵥ nanosheets, visible light-driven CO₂ reduction experiments were carried out under ambient conditions (Supplementary Fig. 35a, b). The circulating cooling water was used to keep the system temperature at around 21 °C according to the in situ thermographic photographs (Supplementary Fig. 35c–f). Xe lamp with VISREF (350–780 nm) reflector, AM 1.5 G filter and 400 nm cutoff filter was used to simulate the visible light source. The corresponding light spectrum is displayed in Supplementary Fig. 36. The instrument was initially evacuated three times, and, afterward, pumped by high-purity CO₂ to reach atmospheric pressure. The gas and liquid products were detected by gas chromatography (GC) and ¹H nuclear magnetic resonance (NMR) spectrum, respectively. As shown in Supplementary Figs. 37-38, GC confirms that the gas products include H₂, CO, CH₄, and C₂H₄, while O₂ is detected as the oxidation product. According to the calibration curves in Supplementary Fig. 39, the corresponding yield of CO, CH₄, and C₂H₄ over CZTS-Sᵥ nanosheet is calculated to be 0.10, 14.33 and 25.16 μmol g⁻¹ h⁻¹ (0.001, 0.1433 and

0.2516 μmol h⁻¹), respectively (Fig. 3d). The turnover number of CZTS-Sᵥ nanosheet for C₂H₄ production in 12 h is 5.64. CMTS, CCTS, and CNTS nanosheets also exhibit considerable performance for visible-light-driven CO₂ reduction to ethylene (Supplementary Fig. 40), with the generation rate of 20.80, 9.05, and 10.18 μmol g⁻¹ h⁻¹, respectively. Experimental results reveal that CZTS and CMTS nanosheets yield higher C₂H₄ production, while CCTS shows the lowest activity, consistent with theoretical predictions (Fig. 1c). This is attributed to the moderate CO* adsorption energies and C–C coupling barriers of CZTS and CMTS, which favor C₂₊ product formation. In contrast, the strong CO* binding on CCTS could increase the CO* coverage but leads to a higher C-C coupling energy barrier, limiting its C₂₊ activity. The ¹H NMR spectrum in Supplementary Figs. 38 and 41 imply that no liquid product is generated for all the synthesized Cu₂MᵢMⱼS₄-Sᵥ nanosheets during the CO₂ reduction. As a comparison, the performance of bulk CZTS was evaluated, as illustrated in Supplementary Fig. 42. Only traces of CH₄ (0.89 μmol g⁻¹ h⁻¹) and CO (0.21 μmol g⁻¹ h⁻¹) were detected during the catalysis. It is notable that the generation rate of O₂ from water splitting is around 93.64 μmol g⁻¹ h⁻¹ (Supplementary Fig. 43) when using the CZTS-Sᵥ nanosheet as the catalyst, confirming that the number of electrons and holes consumed during the catalysis

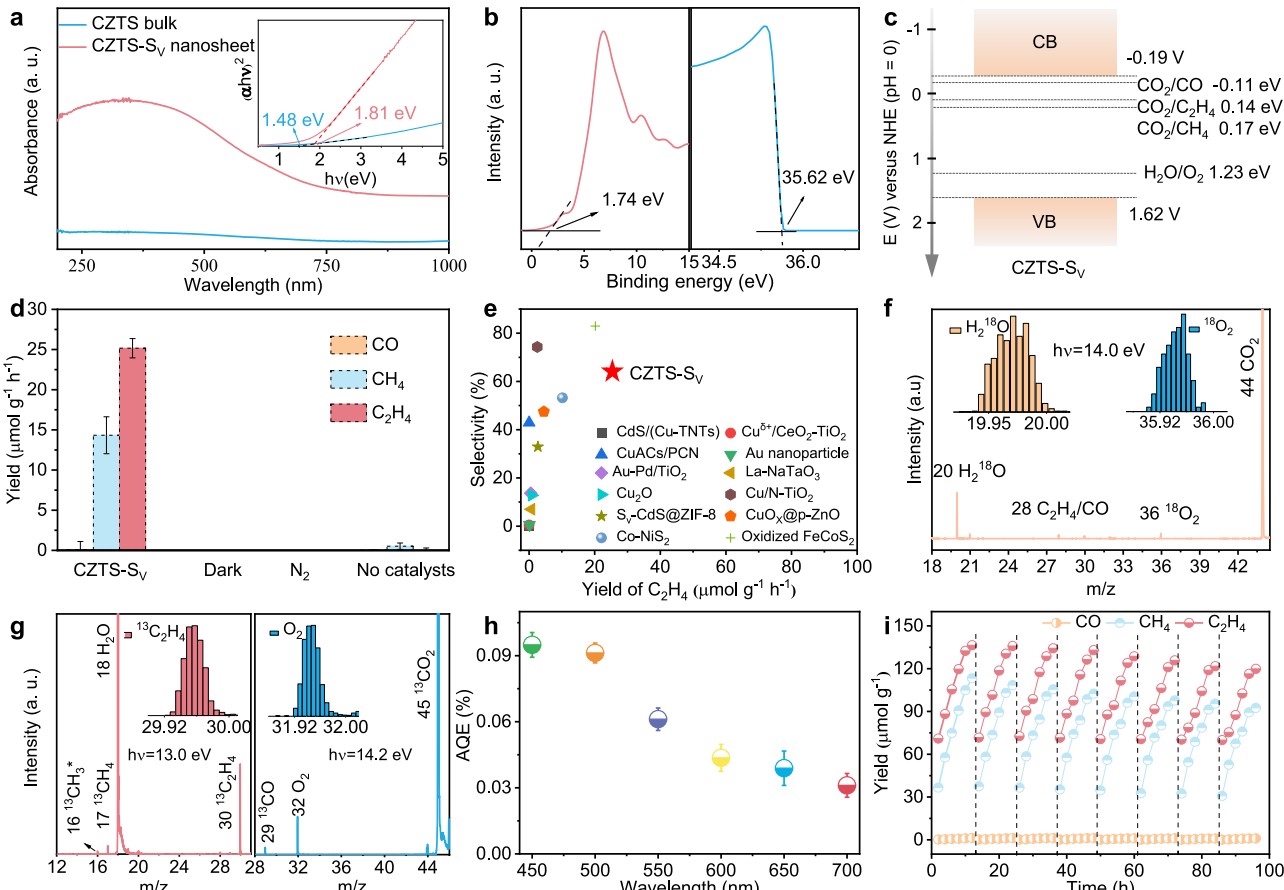

**Fig. 3 | CZTS-S$_v$ nanosheet for visible-light-driven CO$_2$ reduction. a** UV-vis-NIR diffuse reflectance spectra (Insert: the corresponding Tauc plots for CZTS bulk and nanosheets), in which the obtained optical band gap of CZTS bulk and nanosheet is 1.48 and 1.81 eV, respectively. **b** SRPES valence-band (left) and secondary electron cutoff spectra (right) of CZTS-S$_v$ nanosheet. **c** The corresponding band structure of CZTS-S$_v$ nanosheet. **d** The yields of photocatalytic CO$_2$ reduction to CO, CH$_4$, and C$_2$H$_4$ under different conditions, in which error bars represent the standard deviation (s. d.) of three independent measurements using fresh samples for each measurement. **e** The performance comparison of C$_2$H$_4$ in our work with that in the previous literitures (CdS/Cu-TNTs[31], Cu$^{6+}$/CeO$_2$-TiO$_2$[32], CuACs/PCN[33], Au nanoparticle[34], Au-Pd/TiO$_2$[35], La-NaTaO$_3$[36], Cu$_2$O[37], Cu/N-TiO$_2$[38], S$_v$-CdS@ZIF-8[39], CuO$_x$@p-ZnO[40], Co-NiS$_2$[41], Oxidized FeCoS$_2$[42]). **f** Corresponding SVUV-PIMS spectrum of the photocatalytic products after using CO$_2$ and H$_2$$^{18}$O as reactants for CZTS-S$_v$ nanosheet. Note: the signals were collected at hν = 14.0 eV. Insets: signals of m/z = 20 (H$_2$$^{18}$O) and m/z = 36 ($^{18}$O$_2$). **g** Corresponding SVUV-PIMS spectrum of the photocatalytic products after using $^{13}$CO$_2$ and H$_2$O as reactants for CZTS-S$_v$ nanosheet. Note: the signals of $^{13}$CH$_3$*, $^{13}$CH$_4$, H$_2$O, and $^{13}$C$_2$H$_4$ were collected at hν = 13.0 eV; the signals of $^{13}$CO, O$_2$, and $^{13}$CO$_2$ were collected at hν = 14.2 eV. Insets: signals of m/z = 30 ($^{13}$C$_2$H$_4$) and m/z = 32 (O$_2$). **h** Apparent quantum efficiency of CZTS-S$_v$ nanosheet under various monochromatic light (450, 500, 550, 600, 650, 700 nm), in which error bars represent the standard deviation (s.d.) of three independent measurements using fresh samples for each measurement. **i** 8 Cycling measurements for photocatalytic CO$_2$ conversion using CZTS-S$_v$ nanosheet (when a new catalytic cycle begins, the reactor is pumped and refilled with pure CO$_2$), each cycle is performed for 12 h. Source data are provided as a Source Data file.

has basically reached a balance. Compared with the previously reported catalysts[31–42], like FeCoS$_2$ and Cu-ACS/PCN, the CZTS-S$_v$ nanosheet achieves a higher yield for C$_2$H$_4$ evolution under the same reaction condition. The calculated electron and product selectivity of the C$_2$H$_4$ over the CZTS-S$_v$ nanosheet is 72.4% and 63.6%, respectively, both of which outperform most reports under visible light irradiation (Fig. 3e). The corresponding apparent quantum efficiency (AQE) for CO$_2$ conversion was obtained by evaluating the performance of the CZTS-S$_v$ nanosheet under various monochromatic light source, including 450, 500, 550, 600, 650, and 700 nm. As shown in Fig. 3h, the most AQE is calculated to be 0.095% under 450 nm light irradiation. Their changing trends with wavelength are consistent with the light absorption spectrum (Fig. 3a).

The isotope-labeled H$_2$$^{18}$O mass spectrometry was conducted to explore the generation of oxygen, in which the signals were obtained at hν = 14.0 eV. As shown in Fig. 3f, $^{18}$O$_2$ species were detected, confirming the oxygen production from water oxidation. The isotope-labeled $^{13}$CO$_2$ mass spectrometry was further performed to unveil the

source of the C-based products. The photon energy of 13.0 eV was selected for distinguishing the gas products of $^{13}$CH$_4$ and $^{13}$C$_2$H$_4$, while the signals of $^{13}$CO, O$_2$, and $^{13}$CO$_2$ were collected at hν = 14.2 eV according to their absolute photoionization cross sections in Supplementary Fig. 44. As a result, only $^{13}$CO, $^{13}$CH$_4$, and $^{13}$C$_2$H$_4$ species were detected after using the isotope-labeled $^{13}$CO$_2$ as reactants (Fig. 3g), implying that the C-based products indeed originated from the photocatalytic CO$_2$ reduction. Several controlled experiments, e.g., under an N$_2$ atmosphere, in the dark, and without a catalyst, showed almost no product during the catalysis, indicating that light, CO$_2$ reactant, and CZTS-S$_v$ nanosheet catalyst are all necessary for effective CO$_2$ reduction to C$_2$H$_4$. To assess the durability of the CZTS-S$_v$ nanosheet during the photocatalysis, the cycle stability tests were carried out. As illustrated in Fig. 3i, the yield of each product only shows a minor decrease after eight cycles. The good stability of the catalysts was further confirmed by the XRD patterns, TEM images and EDS mapping images as the morphology and crystal structure for the CZTS-S$_v$ nanosheet remained unchanged before and after eight continuous photocatalysis of a total of 96 h (Supplementary Fig. 45). To further analyze whether

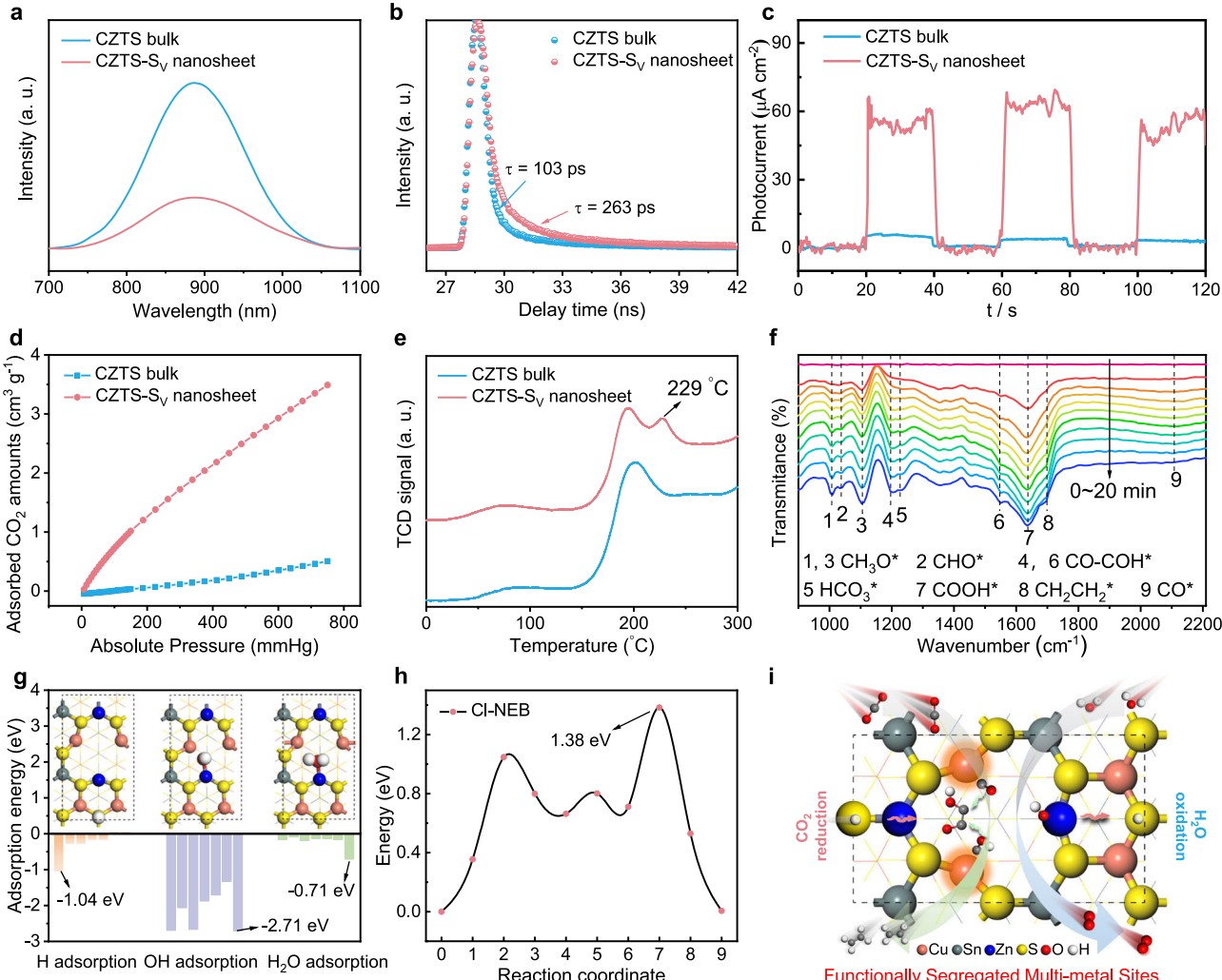

**Fig. 4 | In situ characterization and dynamic exploration of charge carriers.**
**a** Room-temperature PL spectra. **b** Fluorescence emission decay spectra.
**c** Transient photocurrent response spectra. **d** $CO_2$ adsorption isotherms. **e** CO-TPD measurement spectra. **f** In situ FTIR spectroscopy characterization for co-adsorption of a mixture of $CO_2$ and $H_2O$ vapor under light irradiation over CZTS-$S_v$ nanosheet. **g** Adsorption energy of H*, OH* and $H_2O$* intermediates at different sites. **h** The calculated TS energy plots of $H_2O$ splitting to H* and OH* intermediates on CZTS-$S_v$ slab, in which the energy barrier is 1.38 eV. **i** Schematic illustration of a functionally segregated multi-site system for the enhancement of $C_2H_4$ generation efficiency from $CO_2$ conversion, in which Bader charge of Cu sites is regulated by $M_i$, $M_j$ atoms and $S_v$, favoring CO* adsorption and C-C coupling processes; the adjacent S sites promote H* adsorption and transfer (red arrow), collaboratively realizing the $C_2H_4$ production from $CO_2$ and $H_2O$. Source data are provided as a Source Data file.

the samples would be oxidized by the in-situ generated $O_2$ during photocatalysis, we measured the oxygen content on the surface of the samples at different reaction times by TEM EDS mapping. As shown in Supplementary Figs. 46–48, the O content on the CZTS-$S_v$ nanosheet barely increases with the reaction time, confirming its good anti-oxidant properties.

## Mechanistic insights into the $CO_2$ photoreduction processes

To gain mechanistic insights into the processes of $CO_2$ conversion to $C_2H_4$ over the CZTS-$S_v$ nanosheet, the dynamic exploration of charge carriers and the corresponding in situ characterizations were performed. The room-temperature photoluminescence (PL) spectra (Fig. 4a) and the time-resolved fluorescence spectra (Fig. 4b and Supplementary Fig. 49a) illustrate that the CZTS-$S_v$ nanosheet has a lower PL peak intensity and a longer carrier lifetime of 263 ps relative to the bulk counterpart (103 ps), indicating their better separation ability for the excited charge carriers. Meanwhile, the Nyquist plots of electrochemical impedance spectroscopy (EIS) (Supplementary Fig. 49b) and photocurrent density curves (Fig. 4c) demonstrate that the CZTS-$S_v$ nanosheet possesses much smaller resistance for charge

carrier transport, resulting in higher photocurrent density. In addition, compared with CZTS bulk (<1 m² g⁻¹), the CZTS-$S_v$ nanosheet has higher specific surface area (36 m² g⁻¹) according to the Brunauer-Emmet-Teller (BET) plots in Supplementary Fig. 50, which is beneficial for $CO_2$ adsorption, as further confirmed by the $CO_2$ adsorption iso-therms in Fig. 4d. Besides, to verify the better adsorption ability of CZTS-$S_v$ nanosheet for reaction intermediates during the $CO_2$ photo-conversion, temperature-programmed desorption (TPD) spectra for CO gas were performed (Fig. 4e). The desorption peaks at around 198 °C in both CZTS bulk and nanosheets could be ascribed to the CO adsorption at the saturated Cu sites. It is worth noting that there is another desorption peak at around 229 °C for the CZTS-$S_v$ nanosheet, which could be attributed to their stronger adsorption capacity for CO at low-coordination Cu atoms near the S defects in the nanosheets, consistent with the theoretical calculations in Supplementary Fig. 51.

Next, we carried out in situ FTIR measurements to detect the reaction intermediates during $CO_2$ reduction processes for the CZTS-$S_v$ nanosheet (Fig. 4f). With the reaction time increasing from 0 to 20 min, a series of new infrared peaks at around 1641 cm⁻¹ were detected, which are attributed to the COOH* group, a crucial initial

intermediate for $CO_2$ reduction[37]. The absorption bands near 1008 and 1106 cm$^{-1}$ belong to the CH$_3$O* group, and the peaks at 1041 cm$^{-1}$ are assigned to the characteristic bands of CHO*; both the CH$_3$O* and CHO* groups are pivotal intermediates for $CO_2$ reduction to hydrocarbon[43–45].

Moreover, the peaks at 2104 cm$^{-1}$ can be assigned to the adsorbed CO* species[46,47], indicating that the CZTS-S$_v$ nanosheets exhibit strong adsorption for CO* intermediates. The peaks at 1193 and 1547 cm$^{-1}$ belong to the adsorbed CO-COH*[48], consistent with the simulated reaction pathway for $C_2H_4$ generation. Besides, the peaks located at 1697 cm$^{-1}$ are attributed to the $C_2H_4$* group, providing direct evidence for the existence of $C_2H_4$ from the $CO_2$ conversion[49]; and the peaks at 1223 cm$^{-1}$ are inferred to be the asymmetric stretching of the HCO$_3$* group[50]. As suggested by the detected reaction intermediates, the potential reaction pathway for $CO_2$ reduction to $C_2H_4$ over the CZTS-S$_v$ nanosheet can be inferred as follows:

$$*(catalyst) + CO_2 + e^- + H^+ \rightarrow COOH^*(Cu) \qquad (1)$$

$$COOH^*(Cu) + e^- + H^+ \rightarrow CO^*(Cu) + H_2O \qquad (2)$$

$$CO^*(Cu) + CO_2 + e^- + H^+ \rightarrow CO^*(Cu) + COOH^*(Cu) \qquad (3)$$

$$CO^*(Cu) + COOH^*(Cu) + e^- + H^+ \rightarrow CO^*(Cu) + CO^*(Cu) + H_2O \qquad (4)$$

$$CO^*(Cu) + CO^*(Cu) + e^- + H^+ \rightarrow COH^*(Cu) + CO^*(Cu) \qquad (5)$$

$$COH^*(Cu) + CO^*(Cu) \rightarrow COH - CO^*(Cu - Cu) \qquad (6)$$

$$COH - CO^*(Cu - Cu) + e^- + H^+ \rightarrow C - CO^*(Cu - Cu) + H_2O \qquad (7)$$

$$C - CO^*(Cu - Cu) + e^- + H^+ \rightarrow CH - CO^*(Cu - Cu) \qquad (8)$$

$$CH - CO^*(Cu - Cu) + e^- + H^+ \rightarrow CH_2 - CO^*(Cu) \qquad (9)$$

$$CH_2 - CO^*(Cu) + e^- + H^+ \rightarrow CH_2 - CHO^*(Zn) \qquad (10)$$

$$CH_2 - CHO^*(Zn) + e^- + H^+ \rightarrow C_2H_4 + O^*(Zn) \qquad (11)$$

$$O^*(Zn) + e^- + H^+ \rightarrow OH^*(Zn) \qquad (12)$$

$$OH^*(Zn) + e^- + H^+ \rightarrow H_2O + *(catalyst) \qquad (13)$$

In addition to the $CO_2$ reduction reaction, it is important to recognize that the half-reaction of water oxidation also plays a pivotal role in determining the overall conversion efficiency in photocatalysis. Unfortunately, this aspect has often been overlooked in previous reports[51]. Here, we utilized density functional theory (DFT) simulations to explore the possible active sites and reaction mechanisms for $H_2O$ oxidation over the CZTS-S$_v$ slab. As shown in Fig. 4g and Supplementary Fig. 52-54, the calculated adsorption energies of $H_2O$, $H$, and OH species illustrate that the $H_2O$ and OH radicals prefer to bond with Zn atoms near the $S$ defect while the $H$ species are easily adsorbed on the neighboring $S$ atoms, which implies that Zn and $S$ atoms can serve as the active sites for water splitting and H* adsorption storage respectively. To delve deeper into the intricate process of water splitting, CI-NEB TS was carried out as shown in Fig. 4h and Supplementary Fig. 55. The resulting energy barrier for the transition state ($H_2O \rightarrow OH^* + H^*$) was calculated to be 1.38 eV, significantly lower than the TS energy

barrier of the rate-determining step in $CO_2$ reduction (1.695 eV for $CO_2 \rightarrow COOH^*$ as shown in Supplementary Fig. 56). This observation suggests that the milder water-splitting process over the CZTS-S$_v$ slab does not significantly limit the overall efficiency of $CO_2$ and $H_2O$ full splitting.

Combing the theoretical analysis and experimental characterizations, the potential reaction active sites and pathway are proposed as shown in Fig. 4i: $CO_2$ molecule is first activated and reduced to CO* species by COOH* intermediates, while the low-coordination Cu atom pairs can promote the adsorption of CO species and further the C−C coupling process. Moreover, the accelerated water splitting processes by Zn and $S$ sites provide more protons for the production of higher-order hydrocarbons.

## Discussion

In summary, we have demonstrated that both the energy of CO* adsorption and C-C coupling exhibited a linear relationship with the Bader charge of the low-coordination Cu atoms in multi-site $Cu_2M_iM_jS_4$ (M$_i$ = Mn, Fe, Co, Ni, Zn; M$_j$ = Ge, Sn) nanosheets. The Bader charge can serve as a descriptor to predict the performance of $Cu_2M_iM_jS_4$ for $CO_2$ reduction to $C_2H_4$, in which we proposed the BRP principle−apposite Bader charge could provide moderate energy barrier for intermediate adsorption and C−C coupling simultaneously, thus promoting the performance for $C_2H_4$ generation. Experimentally, our designed and synthesized multi-site $Cu_2M_iM_jS_4$-S$_v$ (M$_i$ = Mn, Co, Ni, Zn; M$_j$ = Sn) nanosheets, all of which exhibited an activity for $CO_2$ photoreduction to ethylene. Especially, the CZTS-S$_v$ nanosheet, possessing the most suitable Bader charge among the $Cu_2M_iSnS_4$-S$_v$ nanosheets according to the theoretical prediction, achieved a yield of 25.16 μmol g$^{-1}$ h$^{-1}$ for $C_2H_4$ generation with the electron selectivity of 72.4%, in which the system could be stably operated over 96 h under visible light irradiation. As exemplified by the CZTS-S$_v$ nanosheet, we found that the functionally segregated active sites of Cu, Zn, Sn, and S are responsible for $CO_2$ reduction, $H_2O$ dissociation, H* adsorption storage and transport, respectively, resulting in a mild energy barrier for $CO_2$ conversion to $C_2H_4$. In situ characterizations were further performed to monitor the reaction processes, providing a comprehensive analysis of the catalytic mechanism. We believe that this multi-metal regulation approach and BRP principle can be extended to other alloy compounds and also applied to other catalytic reactions, such as the nitrogen reduction reaction and oxygen reduction reaction.

## Methods

### Materials

Copper powder (99.999% trace metals basis), copper chloride (powder, 99%), manganese nitrate tetrahydrate (≥99.9% trace metals basis), cobalt nitrate hexahydrate (≥99.9% trace metals basis), nickel nitrate hexahydrate (99.999% trace metals basis), zinc powder (99.995% trace metals basis), tin powder (99.99% trace metals basis), sulfur (99.98% trace metals basis), copper acetate monohydrate (≥99%), zinc acetate dehydrate (≥99%), tin chloride pentahydrate (98%), thiourea (ACS reagent, ≥99.0%), polyvinylpyrrolidone (PVP) (average mol wt 40,000) and thioacetamide (≥99%) are all acquired from Sigma-Aldrich and were used without any further purification. Deionized (DI) water with the resistivity of 18.2 MΩ.cm is obtained by the ultra-pure water system from Stakpure GmbH, Germany.

### Catalysts synthesis

**Synthesis of $Cu_2ZnSnS_4$ bulk.** 128 mg Cu powder, 66 mg Zn powder, 112 mg Sn powder, and 128 mg S powder was added into the quartz seal tube. The tube was vacuumed and refilled by Ar for three times. Afterwards, the tube is putted into the high-temperature muffle furnace. $Cu_2ZnSnS_4$ bulk is obtained after calcination at 450 °C for 48 h with the heating rate of 5 °C/min.

**Synthesis of CZTS-S$_v$ nanosheet.** 91 mg copper acetate monohydrate, 55 mg zinc acetate dehydrate, 88 mg tin chloride pentahydrate, and 80 mg thioacetamide were added into 40 mL ethylene glycol. Afterwards, the solution was transferred into a 50 mL Teflon-lined autoclave, sealed and heated at 180 °C for 24 h, and allowed to cool to room temperature naturally. The final product was collected by centrifuging the mixture, washed with ethanol and DI water for several times until the organic residuals were completely removed, and then dried in a vacuum oven at 60 °C overnight. The powder was obtained for further usage.

**Synthesis of CMTS-S$_v$ nanosheet.** 68 mg copper chloride, 62.8 mg manganese nitrate tetrahydrate, and 56 mg tin chloride pentahydrate were added into 40 mL ethylene glycol. After vigorously stirring for 30 min, 70 mg thiourea, and 160 mg PVP were added into the above solution and stirring for another 30 min. Afterwards, the solution was transferred into a 50 mL Teflon-lined autoclave, sealed and heated at 180 °C for 6 h, and allowed to cool to room temperature naturally. The final product was collected by centrifuging the mixture, washed with ethanol and DI water for several times until the organic residuals were completely removed, and then dried in vacuum oven at 60 °C overnight. The powder was obtained for further usage.

**Synthesis of CCTS-S$_v$ nanosheet.** The synthesis process is the same as CMTS-S$_v$ synthesis except that manganese nitrate tetrahydrate is replaced by 73 mg cobalt nitrate hexahydrate.

**Synthesis of CNTS-S$_v$ nanosheet.** The synthesis process is the same as CMTS-S$_v$ synthesis except that manganese nitrate tetrahydrate is replaced by 73 mg nickel nitrate hexahydrate.

## Characterization
XRD patterns were obtained from a Philips X'Pert Pro Super diffractometer with Cu Kα radiation ($\lambda = 1.54178$ Å). TEM and HRTEM images were performed with a JEOL Jem F-200C TEM with an acceleration voltage of 200 kV. UV-vis-NIR diffuse reflectance spectra were measured on a Perkin Elmer Lambda 950 UV-vis-NIR spectrophotometer. XPS spectra were acquired on an ESCALAB MKII system with Al Kα ($h\nu = 1486.6$ eV) as the excitation source. The binding energies obtained in the XPS spectral analysis were corrected for specimen charging by referencing C 1 s to 284.8 eV. In-situ FTIR spectra were obtained by using a Thermo Scientific Nicolet iS50. UPS was performed at the Catalysis and Surface Science Endstation at the BL11U beamline of the National Synchrotron Radiation Laboratory. The workfunction (WF) was determined by the difference between the photon energy and the binding energy of the secondary cutoff edge. To be exact, $E_B = h\nu - (E_K + 4.3 - 5.0)$ and WF $= h\nu - (E_{cutoff} - E_F)$ ($E_B$, binding energy; $h\nu$, photon energy; $E_K$, kinetic energy; $E_{cutoff}$, secondary cutoff edge; $E_F$, Fermi level; photon energy of 40.0 eV and a sample bias of -5 V applied to observe the secondary electron cutoff). Fluorescence emission decay spectra were recorded with a DeltaFlex-NL (HORIBA Scientific) spectrometer. BET isotherms were conducted by 3 Flex Multiport Surface Characterization Analyzer from Micromeritics. Micromeritics ChemiSorb 2720 with a thermal conductivity detector (TCD) were used to conduct TPD of the samples. Room-temperature ESR spectra were measured on an ESR spectrometer (JEOL JES-FA200) at 300 K and 9.062 GHz. Room-temperature PL spectra were acquired on a luminescence spectrometer (Jobin Yvon Fluorolog 3-TAU, Jobin Yvon Instruments), in which the excitation pulse was generated by a 450 W Xe lamp. Synchrotron X-ray absorption spectroscopy (XAS) was performed at P65 beamline at the PETRA III synchrotron (DESY, Hamburg, Germany). XAS data were collected in the transmission mode at room temperature. The incident, transmitted, and reference X-ray intensities were monitored using gas ionization chambers. A Copper foil standard served as a reference for energy calibration and

was concurrently measured with the experimental samples. Prior to measurement, all powder samples were prepared as pellets (diameter: 8 mm), diluted with an appropriate quantity of cellulose, and sealed in Kapton tape. The collected XANES and EXAFS data were further integrated and corrected using Athena software (version 0.9.26).

## In situ FTIR spectra experiments
All FTIR spectra were recorded on Thermo Scientific Nicolet iS50. The spectra were displayed in transmission units and acquired with a resolution of 4 cm$^{-1}$, using 64 scans. The dome of the reaction cell had two KBr windows allowing IR transmission and a third window allowing transmission of irradiation introduced through a liquid light guide that connects to the same IR-light lamp. The catalysts were first added to the reaction cell and then trace amounts of water were sprayed on the surface of catalysts. After degassed in the Ar atmosphere for 20 min, the gas flow was switched to high-purity and wet $CO_2$ until the adsorption was saturated, then the reaction cell was sealed. Next, the FTIR spectra were recorded as a function of time to investigate the dynamics of the reactant adsorption in the dark and desorption/conversion under irradiation[30].

## Photocatalytic $CO_2$ reduction tests under visible light irradiation
For powder catalysts, we used the same method to assess their performance with the previous report[30]. Before performing the $CO_2$ photoreduction performance, we fabricated the sample into a thin film: the sample was dispersed in DI water to gain a concentration of about 5 mg mL$^{-1}$, and then, through spin-dropping 2 mL of the above dispersion on a quartz glass, followed by heat treatment at 65 °C for 30 min, the catalysts thin film could be achieved. During the $CO_2$ photocatalytic process, a MC-PF-300C Xe lamp with VISREF (350−780 nm), AM 1.5 G filter and 400 nm cutoff filter was used to simulate visible light, the corresponding illumination spectrum of which in comparison with sunlight is displayed in Supplementary Fig. 24. Note that the distance from the lamp to the sample was ~10 cm, and the irradiation area of sample is around 7.07 cm$^2$ with an output light density of ~50 mW cm$^{-2}$. The instrument was initially evacuated three times, afterwards, pumped by high-purity $CO_2$ to reach atmospheric pressure. To exclude the heating effect, the homothermal condensate water was used to enable the catalysts to retain a constant temperature of 290 ± 0.2 K. The gas products were quantified by the Agilent GC-8860 gas chromatograph equipped with TDX-01 column, TCD and flame ionization detector (FID) while ultrahigh-purity argon was used as a carrier gas (FID detector for carbon-based products and TCD detector for H$_2$). For O$_2$ detection, Porapak Q 80/100 SS column was used for gaseous oxygen molecules and the dissolved oxygen and other oxidation products were not taken into account. The liquid products were quantified by NMR (Bruker AVANCE AV III 400) spectroscopy, in which dimethyl sulfoxide (DMSO, Sigma, 99.99%) was used as the internal standard.

The product selectivity and electron selectivity for $CO_2$ reduction to ethane has been calculated using the following equation:

$$\text{Product selectivity of } C_2H_4 (\%) = [n(C_2H_4)]/[n(CO) + n(CH_4) + n(C_2H_4)] \times 100\%$$

(14)

$$\text{Electron selectivity of } C_2H_4 (\%) = [12n(C_2H_4)]/[2n(CO) + 8n(CH_4) + 12n(C_2H_4)] \times 100\%$$

(15)

where n(CO), n(CH$_4$) and n(C$_2$H$_4$) are the amounts of produced CO, CH$_4$ and C$_2$H$_4$.

## Photoelectrochemical measurements
Photocurrent and EIS were performed using an electrochemical workstation (CHI 660E, CH Instruments, Shanghai, China).

Specifically, 10 mg of catalysts were dispersed in a mixture containing 950 μL ethanol and 50 μL Nafion solution, then ultrasonic treated to form homogenous catalyst ink. Then the catalyst ink was dipped on a polished FTO glass and dried in air. The photocurrent measurements were conducted in a three-electrode cell system under irradiation of the same light source as that during visible-light-driven $CO_2$ reduction. The FTO glass ($2.5 \times 1.5$ cm$^2$) deposited with materials was used as the photoelectrode, a Pt foil was used as the counter electrode, and Ag/AgCl electrode was used as the reference electrode. The three electrodes were inserted in a quartz cell filled with 0.2 M $Na_2SO_4$ electrolyte. The $Na_2SO_4$ electrolyte was purged with $CO_2$ for 1 h prior to the measurements. EIS was measured in the frequency of 1–1000 kHz[30].

### Apparent quantum efficiency

The wavelength dependent AQE of CZTS-S$_v$ nanosheet for $CO_2$ reduction reaction is defined by the ratio of the effective electrons used for $CO_2$ conversion to the total input photons of different monochromatic light source. Six different light filters of 450, 500, 550, 600, 650, and 700 nm were used to obtain the monochromatic wavelengths (square bandgaps filter with a center wavelength range of ±10 nm). The incident light intensity was determined using a Silicon-UV enhanced actinometer. After 12 h of $CO_2$ reduction, the AQE was estimated from the following equation:

$$AQE\% = \text{Effective electrons}/\text{Total photons} \times 100\% = [e(n) \times Y(n) \times N]/I \times 100\% \quad (16)$$

$$I = (P \times S \times \lambda \times t)/(h \times c) \quad (17)$$

where $Y(n)$ is the yield of different products, including carbon monoxide, methane and ethylene, $e(n)$ is the required electron number for each product, $N$ is Avogadro's number, $P$ is the power of the incident monochromatic light, $I$ is the incident photon number, S is the illumination area, $\lambda$ refers to the wavelength, $t$ is the irradiation time, $h$ corresponds to the Planck constant, $c$ stands for the speed of light.

The following calculation example is based on the data from $CO_2$ photoreduction with CZTS-S$_v$ nanosheets under monochromatic light of 450 nm wavelength for 12 h:

Y(CO) is trace (~ 0), Y(CH$_4$, C$_2$H$_4$) = $4.2 \times 10^{-7}$ mol, $9 \times 10^{-7}$ mol; e(CO, CH$_4$, C$_2$H$_4$) = 2, 8, 12; $N = 6.022 \times 10^{23}$ mol$^{-1}$; $T = 12$ h, $S = 7.07$ cm$^2$; the power of the incident 450 nm monochromatic light: 13 mW cm$^{-2}$.

$$AQE\% = [(8 \times 4.2 \times 10^{-7} + 12 \times 9 \times 10^{-7}) \times 6.022 \times 10^{23}]$$
$$(6.626 \times 10^{-34} \times 3 \times 10^8)/(13 \times 7.07 \times 10^{-3} \times 450 \times 10^{-9} \times 12 \times 3600) = 0.095\%.$$

### DFT calculation details

DFT calculations were carried out on a Vienna Ab initio Simulation Package[52]. The exchange-correlation potential was described by the generalized gradient approximation within the framework of Perdew-Burke-Ernzerhof (PBE) functional[53]. DFT-D3 method was employed to calculate the van der Waals (vdW) interaction[54]. The parameters of dipole correction were applied for the calculation of slab models. Electronic energies were computed with the tolerance of $1 \times 10^{-5}$ eV and total force of 0.01 eV/Å. A kinetic cutoff energy of 450 eV was adopted. The optimized crystal lattice parameters of CZTS bulk (space group (I-42m (121))) are as follows: a = b = 5.4348 Å, c = 10.9589 Å ($\alpha = \beta = \gamma = 90°$). A Monkhorst-Pack k-mesh of $5 \times 5 \times 3$ k-points were used in the structural relaxation for the bulk. The $Cu_2ZnSnS_4$ slabs were modeled by the corresponding exposed surface along (001), (100), and (112) direction with the thickness of one unit cells, in which two bottom atoms are fixed to simulate the bulk structure. A

Monkhorst-Pack k-mesh of $4 \times 2 \times 1$ k-points were used in the structural relaxation for slab models. A vacuum space of 15 Å was inserted in z direction to avoid interactions between periodic images. (More details see Supplementary Notes).

The surface energy $E_s$ is the energy required to cleave a surface from the corresponding bulk crystal. It can be given by

$$E_s = 1/2 A [E_s(\text{unrelax}) \cdot N \times E_b] + 1/A [E_s(\text{relax}) - E_s(\text{unrelax})] \quad (18)$$

where A is the area of the surface on the slab models, $E_s$(unrelax) and $E_s$(unrelax) represent the energy of the unrelaxed and relaxed surface slab models, respectively. N is the number of in the slab and $E_b$ is the energy of each atom in the bulk counterpart[30].

Ab initio molecular dynamics simulations are carried out within the canonical (NVT) ensemble at 300 K with the time step of 3 fs, in which at least 10 ps is used to allow the structure to relax and reach the relative equilibrium state.

Adsorption energies $E_{adsorption}$ are given with reference to the isolated surface $E_{surface}$ relaxed upon removing the molecule from the unit cell using identical computational parameters and the energy of the molecule $E_{molecule}$[30].

$$E_{adsorption} = E_{molecule\ on\ surface} - E_{surface} - E_{molecule} \quad (19)$$

The computational hydrogen electrode[55] model was used to calculate the Gibbs free energy change (ΔG) of $CO_2$ reduction reaction steps:

$$G = E_{DFT} + E_{ZPE} - TS \quad (20)$$

$$E_{ZPE} = \sum_i 1/2\, h\nu_i \quad (21)$$

$$\Theta_i = h\nu_i/k \quad (22)$$

$$S = \sum_i R[\ln(1 - e^{-\Theta_i/T})^{-1} + \Theta_i/T(e^{\Theta_i/T} - 1)^{-1}] \quad (23)$$

where $E_{DFT}$ is the electronic energy calculated for specified geometrical structures, $E_{ZPE}$ is the zero-point energy, S is the entropy, h is the Planck constant, ν is the computed vibrational frequencies, Θ is the characteristic temperature of vibration, k is the Boltzmann constant, and R is the molar gas constant. For adsorbates, all 3 N degrees of freedom were treated as frustrated harmonic vibrations with negligible contributions from the catalysts' surfaces[30].

The CI-NEB method is used to evaluate the energy barriers of transition states during C–C coupling and $H_2O$ dissociation[56,57]. The image dependent pair potential method was used for improved interpolation of NEB initial guess[58]. The C–C coupling and $H_2O$ dissociation processes can be represented by

$$COH^* + CO^* \rightarrow COH - CO^* \quad (24)$$

$$H_2O^* \rightarrow H^* + OH^* \quad (25)$$

For the theoretical energy band structure, the screened hybrid functional proposed by Heyd, Scuseria, and Ernzerhof[59] was adopted to precisely calculate the DOS, from which band gap of CZTS can be obtained. The surface electrostatic potential is also computed to gain the work functions. Combining the DOS and work function, we can illustrate their band structure in theory.

## Data availability

The data that support the plots within this paper and other findings of this study are available from the corresponding author upon reasonable request. Source data are provided with this paper.

## Code availability

The code that supports the findings of this study is available from the corresponding author upon request.

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

## Acknowledgements
This work was financially supported by the European Research Council (ERC) under the European Union's Horizon 2020 research and innovation program (grant agreement No 819698 and GrapheneCore3: 881603), Deutsche Forschungsgemeinschaft (COORNETs, SPP 1928 and CRC 1415: 417590517). National Natural Science Foundation of China (22125503). The Supercomputing Center of Max Planck Computing & Data Facility (MPCDF) is acknowledged for computational support.

## Author contributions
X.F., Y.S., and X.D.L. conceived the idea and co-wrote the paper. X.D.L., L.L., X.H.L., X.C., G.C., M.W., D.L., X.W., and J.G. carried out the sample synthesis, characterization and $CO_2$ reduction measurement. X.D.L., L.L., X.H.L., J.X., and M.G. discussed the catalytic process. C.N. helped with the BET characterizations. All the authors contributed to the overall scientific interpretation and edited the manuscript.

## Funding

## Competing interests
The authors declare no competing interests.
