## [Peer Review File · Nature Communications]

Design of Multi-Metal-Site Nanosheet Catalyst for Enhanced CO₂ Photoreduction to Ethylene

Corresponding Author: Professor Xinliang Feng

Version 0:

Reviewer comments:

Reviewer #1

(Remarks to the Author)

In this manuscript, Li and co-workers proposed a novel principle to describe the structure-activity relationship between kesterite-like Cu₂MiMjS₄ nanosheets and their performance for CO₂ photoreduction to ethylene. And according to the theoretical prediction, they also synthesized a series of Cu₂MiMjS₄ nanosheets. Among them, the obtained Cu₂ZnSnS₄ (CZTS) nanosheet achieved a record ethylene yield under visible light irradiation. They also explored the catalytic reaction process and mechanism through in situ characterizations, isotope labeling experiments, etc. Their findings are super interesting, and the experimental phenomena are in good agreement with the conclusions. Based on the content of the work, I think the novelty and significance of this manuscript are suitable for publication in "Nature communications". The following comments and suggestions should be addressed.

1. The authors took the CZTS bulk as the comparison, is it possible to synthesize CZTS nanosheet without S vacancies? If yes, it should be better to use the CZTS nanosheet as the comparison so that they can only consider the effect of S vacancies on the performance.
2. Is it possible to regulate the concentration of the S vacancies in the sample? I noticed that the author didn't define the amount of the S vacancies.
3. How about the scalability of this method for synthesizing CZTS nanosheet?
4. Considering the good performance of CZTS nanosheet for C₂H₄ production, what is the cost and practical application possibility of the catalyst?
5. Oxygen is produced during the photocatalytic reaction. Will the catalyst be oxidized?

Reviewer #2

(Remarks to the Author)

In this manuscript, Li et al. provided a novel approach to designing multi-metal-site nanosheet catalysts for efficient CO₂ photoreduction to C₂H₄. A Bader-Regulate-Performance (BRP) principle to balance CO* adsorption and C-C coupling energy barrier was proposed. The Cu₂ZnSnS₄-Sv nanosheet achieves a record ethylene yield. The work excels in bridging computational insights with practical catalyst design, offering a robust framework for optimizing multi-carbon product formation. Therefore, the work is highly original and the proposed methodology, and achieved results are rigorous. I would recommend it for publication in Nature Communications. Here are several suggestions to improve the work further.

1. How O₂ evolution rates were measured (e.g., via GC or iodometric titration), as this data is critical for mass/charge balance but only briefly mentioned in Supplementary Figure 41. Detailed information should be added to the methods.
2. Related literatures should be cited, such as Small, 2022,18, 2203759; Nature Communications 2023, 14, 1298; Nature Reviews Chemistry, 2024, 8, 410.
3. In Figure 1c, the absence of a legend makes it unclear what the red and blue colors signify. It is crucial to include a legend to enhance figure readability and understanding.
3. The manuscript inconsistently uses chemical terms such as ZnSe and CoSn, and in other sections, equivalent terms like CZTS and CCTS are used. It is recommended to standardize these expressions throughout the text to avoid confusion and

maintain consistency.

- The text suddenly introduces 'Sv' (sulfur vacancies) without clear context or explanation of its impact on reaction activity ("We found that the Sv can be spontaneously generated on the surface of....."). It is advisable to provide a preliminary discussion on the role and effects of Sv early in the manuscript, particularly before detailed discussions or data presentations that involve Sv.
- The manuscript identifies the elementary step of $\text{CO}_2 \rightarrow \text{COOH}^*$ as the rate-determining step (RDS) based on Gibbs free energy calculations. However, the significance of investigating other reaction steps remains unexplained. The text should clarify why it is important to study these additional steps if $\text{CO}_2 \rightarrow \text{COOH}^*$ is already determined as the RDS.
- Could the manuscript highlight potential discrepancies between theoretical predictions (e.g., Bader charge trends) and experimental results (e.g., the activity of different catalysts in Supplementary Figure 38 nanosheets) to strengthen validation?
- The H_2O splitting and H transfer should also be included in Fig. 4i.
- The in-text citations from ref. 23 should be revised to match the numbered reference list in sequence.
- Many photocatalytic reports involve dispersing the catalyst in a solution. Why did the authors use a gas-solid method? Are there any advantages?

Reviewer #3

(Remarks to the Author)

In this manuscript, Feng et al propose the Bader-Regulate-Performance principle apposite Bader charge to provide a moderate energy barrier for intermediate adsorption and C-C coupling simultaneously, thus promoting the performance of C=C bonded ethylene generation. The proposed nice concept has been carefully elucidated with modeling and in-situ experimental characterizations, which is of great interest to the photocatalytic CO_2 reduction research community. I enjoy reading this manuscript. Herein, I suggest further understanding the structural and C=C bond formation mechanism carefully before its acceptance. Detailed suggestions for further improvement of the manuscript have been given below:

- As it has been identified that Mi and Mj atoms, along with S vacancies, help regulate the Bader charge of Cu atoms and facilitate water decomposition, what is the optimal combination of Mi and Mj within this structural matrix? Are there any other functions of these two metal positions beyond the Bader charge regulation?
- The bimetal semiconductor photocatalyst has been extensively studied to synthesize hydrocarbon compounds via C-O coupling (Nature Communications, 2022, 13: 2146) or C-C coupling (Nano Lett. 2023, 23, 23, 10914–10921), or simply C-H bonded methane (Angew. Chem., Int. Ed., 2023, 62, 39, e202309625), most of the reported cases, there just saturated single bonds within the molecular, in this case, the double C=C bond involved unsaturated bonds have been reported, what are the driving forces to preserve the double C=C bonds within the reduced products? Could the author elaborate on this fascinating phenomenon in more detail and provide some suggestions for constructing other effective photocatalysts to produce products containing unsaturated bonds?
- By the way, the synergistic effects of different metal centers in selectively achieving C=C double bond products or other types of bonding products need to be discussed within the revised manuscript.
- What are the space groups and unit cells of the as-synthesized four different composition materials? Why have their high-resolution image been indexed to different crystallographic planes while preserving the same interplanar angle?
- By the way, are the observed different nanosheet orientations directly related to the calculation model?
- How do you consider the atomicity within the different crystal structural models used in calculation compared to the experimental ones? From the structure models shown in the manuscript, both Mi and Mj have well-defined positions; is that consistent with the experimental data? Furthermore, I suggest using different colors to represent Mi and Mj; currently, both are represented by grey, which makes it challenging for readers to distinguish them from the figures.
- When writing the redundant CO_2 on both sides of Equations 1 and 2? It may be challenging for readers unfamiliar with this area to understand these concepts. Is it also beneficial to include the metal centers that absorb these molecules within the equations? Similar problems also happen for other intermediates.
- The author claimed that sulfur atoms act as an active site for hydrogen adsorption; does any in-situ spectroscopy data support this claim rather than calculation?

Reviewer #4

(Remarks to the Author)

This manuscript proposed the Bader-Regulate-Performance principle — apposite Bader charge can provide a moderate energy barrier for intermediate adsorption and C-C coupling simultaneously, thus promoting the performance for ethylene generation from CO_2 photoreduction, specifically focusing on the $\text{Cu}_2\text{MiMjS}_4$ (Mi=Mn, Fe, Co, Ni, Zn; Mj=Ge, Sn) nanosheets. The $\text{Cu}_2\text{ZnSnS}_4$, with the appropriate Bader charge, achieves a record ethylene yield under visible light irradiation. Although the theoretical expectation is interesting, several limitation should be considered.

- In the mechanistic study of the C-C coupling pathway (including the three typical pathways: CO-CO, CO-CHO, and CO-COH), the authors did not provide activation energy barrier calculations for the direct coupling of the CO dimer (CO-CO). The current analysis is based solely on the energy barrier for the formation of the COCOH intermediate via CO and COH, making it difficult to accurately assess the C-C coupling mechanism. It is recommended to supplement the bond formation energy barrier of $^*\text{CO}-^*\text{CO}$ coupling and compare it with the hydrogenation-protonation process of $^*\text{CO}-^*\text{CO}$ (as these two steps are competitive), to more cautiously determine the most probable coupling pathway and subsequent selection criteria.

2. In the manuscript, the hydrogenation of CO₂ is identified as the rate-determining step (RDS) based on the thermodynamic energy barrier, which is inappropriate. Moreover, the free energy barrier for this process is 1.52 eV, indicating that this step is thermodynamically challenging. However, experimental results show that the CZTS-Sv catalyst exhibits excellent catalytic performance for CO₂ reduction to ethylene, which contradicts the theoretical prediction. It is recommended to verify this discrepancy.

3. In lines 312-315 on page 10 of the main text, the authors directly compare the transition state energy barrier for H₂O dissociation (1.38 eV) with the free energy barrier for CO₂ hydrogenation to form the *COOH intermediate (1.52 eV). This approach presents an inconsistency in the evaluation criteria. In fact, the energy barrier comparison for competing reaction pathways should follow one of the following principles: (i) using transition state energy barriers consistently (for kinetically controlled systems), or (ii) using free energy barriers consistently (for thermodynamically controlled systems). It is recommended to recalculate the transition state energy barrier for the CO₂ → *COOH pathway or to use free energy changes uniformly for a cross-comparison to accurately assess the impact of the hydrolysis reaction on CO₂ reduction.

4. In the theoretical screening section of Figure 1c, the authors did not clearly explain the performance advantage criteria for CZTS-Sv within the orange region (e.g., adsorption energy, coupling energy barrier, and other key indicators). Although the manuscript mentions that catalysts in this region exhibit superior properties, the lack of well-defined screening benchmarks weakens the persuasiveness of the theoretical guidance. Furthermore, the experimental section characterizes and tests five catalysts within the screening range, exposing the limitations of the theoretical predictions.

Version 1:

Reviewer comments:

Reviewer #1

(Remarks to the Author)

This revised manuscript has addressed all the points raised by reviewers and is now recommended for publication as it is.

Reviewer #2

(Remarks to the Author)

The author addressed all my comments, I would recommend it publish, thanks.

Reviewer #3

(Remarks to the Author)

My concern has been well addressed within the revised manuscript. It can be accepted for publication. Congratulations to the authors!

Reviewer #4

(Remarks to the Author)

The authors have addressed the reviewers' concerns well, and the manuscript has also been improved much. Thus, the paper can be accepted for publication as it is.

Responses to the Reviewers' comments and a summary of the changes made to the manuscript: NCOMMS-25-17898-T. We would like to thank all the reviewers for the insightful comments and suggestions, and for their time in helping us to improve this manuscript.

And we acknowledge the Reviewers' positive comments that "*I think the novelty and significance of this manuscript are suitable for publication in Nature communications*", "*I would recommend it for publication in Nature Communications*", "*I enjoy reading this manuscript*" and "*the theoretical expectation is interesting*".

Point-to-point Response to Reviewer #1

Overall comments: *The In this manuscript, Li and co-workers proposed a novel principle to describe the structure-activity relationship between kesterite-like $\text{Cu}_2\text{MiMjS}_4$ nanosheets and their performance for CO_2 photoreduction to ethylene. And according to the theoretical prediction, they also synthesized a series of $\text{Cu}_2\text{MiMjS}_4$ nanosheets. Among them, the obtained $\text{Cu}_2\text{ZnSnS}_4$ (CZTS) nanosheet achieved a record ethylene yield under visible light irradiation. They also explored the catalytic reaction process and mechanism through in situ characterizations, isotope labeling experiments, etc. Their findings are super interesting, and the experimental phenomena are in good agreement with the conclusions. Based on the content of the work, I think the novelty and significance of this manuscript are suitable for publication in "Nature communications".*

Overall response: We greatly appreciate the Reviewer's positive comments and constructive suggestions guiding the revision of our manuscript.

Comment 1: *The authors took the CZTS bulk as the comparison, is it possible to synthesize CZTS nanosheet without S vacancies? If yes, it should be better to use the CZTS nanosheet as the comparison so that they can only consider the effect of S vacancies on the performance.*

Response: We appreciate the Reviewer's suggestions. According to the calculated results in Fig. 1a, the formation energy of S vacancies in CZTS nanosheets is negative, indicating that the formation of S vacancies is a spontaneous process and relatively easy. In the experiment, we also attempted to synthesize CZTS nanosheets under conditions of an excess sulfur source. We found that even when using twice the amount of sulfur source, the synthesized CZTS

nanosheets still exhibited S vacancies, as confirmed by the ESR results in Fig. N1. Therefore, it is difficult to synthesize CZTS nanosheets without S vacancies.

Fig. N1 | ESR spectra for the synthesized CZTS nanosheets using double the sulfur source. The signal at $g = 2.003$ corresponds to the S vacancy.

Comment 2: *Is it possible to regulate the concentration of the S vacancies in the sample? I noticed that the author didn't define the amount of the S vacancies.*

Response: We appreciate the Reviewer's comments. In fact, it is quite difficult to precisely measure and control the concentration of S vacancies. In this work, our focus was solely on investigating the structure-activity relationship between the formation of S vacancies, the regulation of Bader charge on Cu by multi-metal sites, and the promotion of C-C coupling and multi-carbon product formation. We did not study the effect of S vacancy concentration, nor did we attempt to control its content. However, this is a very good suggestion, and we plan to pursue this direction in future work by exploring the influence of S vacancy concentration.

Comment 3: *How about the scalability of this method for synthesizing CZTS nanosheet?*

Response: Following the reviewer's suggestion, we **conducted** scale-up experiments of the synthesis reaction. Using a 100 mL reactor and doubling the amount of reactants (Supplementary Note 4), we successfully synthesized CZTS nanosheets with morphology and size essentially identical to those obtained in previous experiments, as shown in Fig. N2a. Additionally, by employing multiple synthesis reactors, we were able to produce approximately

2 grams of the sample in a single batch (Fig. N2b), confirming the scalability of this method. We **have included** the unscalable synthesis results into the revised Supplementary Information (SI), as shown in Supplementary Figure 31c.

Fig. N2 | (a) TEM image of CZTS nanosheets synthesized under scale-up conditions. (b) The digital image of upscalable synthesis of CZTS nanosheets. Samples of more than 2 grams can be easily synthesized using the solvothermal method.

Comment 4: *Considering the good performance of CZTS nanosheet for C_2H_4 production, what is the cost and practical application possibility of the catalyst?*

Response: We appreciate the Reviewer's comments and suggestions. We **have added** the discussion about the cost and the practical application possibility of the catalyst in the Supplementary Information as follows:

Supplementary Note 4: Cost analysis for synthesizing CZTS-S_v nanosheets:

Chemical	Price	Company (Part Number)
Copper(II) acetate monohydrate	0.1576 € /g	Sigma-Aldrich (1027109050-50KG)
Zinc acetate dihydrate	0.2044 € /g	Sigma-Aldrich (383058-2.5KG)
Tin(IV) chloride pentahydrate	0.217 € /g	Sigma-Aldrich (244678-1KG)
Thioacetamide	0.206 € /g	Sigma-Aldrich (172502-500G)

Ethylene glycol	51.8 € /L	Sigma-Aldrich (102466-5L)
-----------------	-----------	---------------------------

The usage of reactants includes 91 mg copper acetate monohydrate, 55 mg zinc acetate dehydrate, 88 mg tin chloride pentahydrate, 80 mg thioacetamide and 40 mL ethylene glycol. The production of CZTS-S_v nanosheets is approximately 80 mg, and the price is 26.66 €/g, based on the reactant prices. Compared to precious metals, such as Au (90.04 €/g), the price of the obtained sample is more than three times lower, indicating the sustainability of our catalyst.

Comment 5: *Oxygen is produced during the photocatalytic reaction. Will the catalyst be oxidized?*

Response: Thanks for the Reviewer's concerns. We agree that oxygen is produced during the photocatalytic reaction as an oxidation product. However, our catalyst (CZTS nanosheets) has been demonstrated to be stable during photocatalysis. We first carried out the characterizations for the CZTS-S_v nanosheets before and after photocatalysis as displayed in Supplementary Figure 45. The crystal phase, morphology, and element distribution of the catalyst after photocatalysis remained unchanged as before photocatalysis, confirming the stability of the CZTS-S_v nanosheets. Furthermore, we specifically characterized the oxygen content of the catalyst during the catalytic process using TEM-EDS mapping. As shown in Supplementary Figures 46-48, the O content increased only slightly, from 3.8% to 4.0% after 12 hours of photocatalysis, which could be attributed to the adsorption of O₂ or H₂O. Therefore, we concluded that despite the generation of oxygen, the CZTS-S_v nanosheets remain stable during the catalytic process and is almost unaffected by oxidation.

Point-to-point Response to Reviewer #2

Overall comments: *In this manuscript, Li et al. provided a novel approach to designing multi-metal-site nanosheet catalysts for efficient CO₂ photoreduction to C₂H₄. A Bader-Regulate-Performance (BRP) principle to balance CO* adsorption and C-C coupling energy barrier was proposed. The Cu₂ZnSnS₄-S_v nanosheet achieves a record ethylene yield. The work excels in bridging computational insights with practical catalyst design, offering a robust framework for optimizing multi-carbon product formation. Therefore, the work is highly original and the*

proposed methodology, and achieved results are rigorous. I would recommend it for publication in Nature Communications. Here are several suggestions to improve the work further.

Overall response: We greatly appreciate the Reviewer's positive comments and kind suggestions.

Comment 1: *How O₂ evolution rates were measured (e.g., via GC or iodometric titration), as this data is critical for mass/charge balance but only briefly mentioned in Supplementary Figure 41. Detailed information should be added to the methods.*

Response: We appreciate the Reviewer's comments and suggestions. We **have added** the details of the O₂ detection in the Method section. We also explained that due to method limitations, we only quantified the generated gaseous oxygen molecules, and did not take dissolved oxygen and other oxidation products into account.

Comment 2: *Related literatures should be cited, such as Small, 2022,18, 2203759; Nature Communications 2023, 14, 1298; Nature Reviews Chemistry, 2024, 8, 410.*

Response: According to the Reviewer's suggestion, we **have cited** the related literature in the Introduction part of the revised manuscript.

Comment 3: *In Figure 1c, the absence of a legend makes it unclear what the red and blue colors signify. It is crucial to include a legend to enhance figure readability and understanding.*

Response: Thanks for the Reviewer's comments. We **have added** the corresponding legend in revised Figure 1c to clarify the information.

Comment 4: *The manuscript inconsistently uses chemical terms such as ZnSe and CoSn, and in other sections, equivalent terms like CZTS and CCTS are used. It is recommended to standardize these expressions throughout the text to avoid confusion and maintain consistency.*

Response: Thanks for the Reviewer's suggestion. We **have standardized** these expressions for catalysts to CZGS, CZTS, CMTS, CNTS, CCTS, CFTS in the revised manuscript.

Comment 5: *The text suddenly introduces 'Sv' (sulfur vacancies) without clear context or explanation of its impact on reaction activity ("We found that the Sv can be spontaneously generated on the surface of....."). It is advisable to provide a preliminary discussion on the role and effects of Sv early in the manuscript, particularly before detailed discussions or data presentations that involve Sv.*

Response: Thanks for the Reviewer's useful comments. We **have added** a discussion about the effect of sulfur vacancies (Sv) on reaction performance before the data presentations involving Sv in the revised manuscript on page 4 as follows:

"Considering that anion vacancies often lead to local charge accumulation, which can stabilize the reaction intermediates, thereby enhancing catalytic activity and regulating product selectivity,²⁰ we further calculated the formation energy of surface sulfur vacancies."

Comment 6: *The manuscript identifies the elementary step of $\text{CO}_2 \rightarrow \text{COOH}^*$ as the rate-determining step (RDS) based on Gibbs free energy calculations. However, the significance of investigating other reaction steps remains unexplained. The text should clarify why it is important to study these additional steps if $\text{CO}_2 \rightarrow \text{COOH}^*$ is already determined as the RDS.*

Response: Thanks for the Reviewer's concerns. $\text{CO}_2 \rightarrow \text{COOH}^*$ is identified as the rate-determining step (RDS) only after we conducted the comprehensive calculations for C_2H_4 production, which determines the overall reaction's thermodynamic energy barrier. The additional steps, including the C-C coupling process and the C_2H_4 formation pathways, are incorporated to evaluate the feasibility of producing multi-carbon products and the potential for C_2H_4 generation. We already have corresponding statements in the original manuscript: *"The comprehensive calculations of Gibbs free energy for CO_2 reduction to C_2H_4 were conducted to unveil the rate-determining step (RDS) and dynamic conversion processes."*

To present the computational results more clearly, we **have included** additional descriptions as follows in the revised manuscript on page 6:

"That means the thermodynamic energy barrier for C_2H_4 production is induced by the protonation of CO_2 to COOH^ , and the C-C coupling process is not the determining factor while relatively easier to proceed on the CZTS- S_v slab."*

Comment 7: *Could the manuscript highlight potential discrepancies between theoretical predictions (e.g., Bader charge trends) and experimental results (e.g., the activity of different catalysts in Supplementary Figure 38 nanosheets) to strengthen validation?*

Response: Thanks for the Reviewer's suggestion. We **have added** the corresponding explanation to strengthen validation for the theoretical results in the revised manuscript on page 9 as follows:

“Experimental results reveal that CZTS and CMTS nanosheets yield higher C₂H₄ production, while CCTS shows the lowest activity, consistent with theoretical predictions (Fig. 1c). This is attributed to the moderate CO adsorption energies and C-C coupling barriers of CZTS and CMTS, which favor C₂₊ product formation. In contrast, the strong CO* binding on CCTS could increase the CO* coverage but leads to a higher C-C coupling energy barrier, limiting its C₂₊ activity.”*

Comment 8: *The H₂O splitting and H transfer should also be included in Fig. 4i.*

Response: Thanks for the Reviewer's concerns. We already included the H₂O splitting and H transfer processes in Fig. 4i. To make it clearer, we **have changed** the arrows to red to highlight these processes.

Comment 9: *The in-text citations from ref. 23 should be revised to match the numbered reference list in sequence.*

Response: We **have rechecked** the manuscript to ensure all references are in order and correctly matched.

Comment 10: *Many photocatalytic reports involve dispersing the catalyst in a solution. Why does the author use a gas-solid method? Are there any advantages?*

Response: Thanks for Reviewer's concerns. It is well known that the hydrogen evolution reaction (HER) competes with CO₂ reduction reaction. When the catalyst is in direct contact with liquid water (Fig. N3a), HER is more likely to occur, which suppresses the CO₂ reduction reaction. Moreover, the limited solubility of CO₂ in water further reduces its concentration at the catalyst surface, thereby lowering the activity of CO₂ reduction reaction. In contrast, gas-

solid reactions use water vapor as the proton source (Fig. N3b), minimizing contact with liquid water and thus enhancing CO₂ reduction performance while suppressing HER. Additionally, gas-solid systems facilitate easier catalyst recovery without the need for centrifugation, reducing operational costs in practical applications.

Fig. N3 | Schematic diagram of (a) liquid-solid and gas-solid set up for CO₂ photoreduction.

Point-to-point Response to Reviewer #3

Overall comments: *In this manuscript, Feng et al propose the Bader-Regulate-Performance principle apposite Bader charge to provide a moderate energy barrier for intermediate adsorption and C-C coupling simultaneously, thus promoting the performance of C=C bonded ethylene generation. The proposed nice concept has been carefully elucidated with modeling and in-situ experimental characterizations, which is of great interest to the photocatalytic CO₂ reduction research community. I enjoy reading this manuscript. Herein, I suggest further understanding the structural and C=C bond formation mechanism carefully before its acceptance.*

Overall response: We greatly appreciate the Reviewer's positive comments and kind suggestions.

Comment 1: *As it has been identified that Mi and Mj atoms, along with S vacancies, help regulate the Bader charge of Cu atoms and facilitate water decomposition, what is the optimal combination of Mi and Mj within this structural matrix? Are there any other functions of these two metal positions beyond the Bader charge regulation?*

Response: Thanks for the Reviewer's concerns. From the theoretical results in Fig. 1c, we can see that CZTS and CMTS possess moderate CO* adsorption energy for providing higher CO* coverage and mild C-C coupling energy barrier for C₂₊ production, which means CZTS and CMTS could show better performance for CO₂ reduction. Meanwhile, Fig. 1a shows that the S vacancies are more likely to form in CZTS, inducing more unsaturated Cu pairs for CO₂ reduction. Therefore, the combination of Zn and Sn within this structural matrix is optimal. Subsequently, we confirmed this prediction by evaluating their performance for CO₂ photoreduction. As shown in Fig. 3d and Supplementary Figure 40, the CZTS-Sv nanosheets exhibited the best performance for C₂H₄ production, among the Cu₂M_iM_jS₄-S_v nanosheets. To make the manuscript clearer, we **have added** the following description in the revised manuscript on page 9:

“Experimental results reveal that CZTS and CMTS nanosheets yield higher C₂H₄ production, while CCTS shows the lowest activity, consistent with theoretical predictions (Fig. 1c). This is attributed to the moderate CO adsorption energies and C-C coupling barriers of CZTS and CMTS, which favor C₂₊ product formation. In contrast, the strong CO* binding on CCTS could increase the CO* coverage but leads to a higher C-C coupling energy barrier, limiting its C₂₊ activity.”*

Regarding whether there are any other functions beyond the Bader charge regulation, we can't say that for sure at this stage. The electronegativity of M_i and M_j as well as the distance of the unsaturated Cu pairs, could also be factors in determining the performance of Cu₂M_iM_jS₄-S_v nanosheets for CO₂ photoreduction; however, we are still working on this aspect.

Comment 2: *The bimetal semiconductor photocatalyst has been extensively studied to synthesize hydrocarbon compounds via C-O coupling (Nature Communications, 2022, 13: 2146) or C-C coupling (Nano Lett. 2023, 23, 23, 10914-10921), or simply C-H bonded methane (Angew. Chem., Int. Ed., 2023, 62, 39, e202309625), most of the reported cases, there just saturated single bonds within the molecular; in this case, the double C=C bond involved unsaturated bonds have been reported, what are the driving forces to preserve the double C=C bonds within the reduced products? Could the author elaborate on this fascinating phenomenon in more detail and provide some suggestions for constructing other effective photocatalysts to*

produce products containing unsaturated bonds?

Response: Thanks for the Reviewer's concerns. This is really a good question. We **have cited** the mentioned literatures to make our manuscript more comprehensive. Regarding the driving forces to preserve the C=C double bonds, we believe it originates from three factors: First, a suitable CO* adsorption energy, which can prevent CO* from desorbing as free CO gas, thereby increasing the local concentration of CO* intermediates, beneficial for C-C coupling (*Nat. Catal.* 2018, 1, 946); Second, a lower energy barrier for C-C coupling favors the formation of multi-carbon products rather than single-carbon products such as CH₄ or methanol (*Nat. Commun.* 2024, 15, 7053); Third, an appropriate reduction activity — possibly related to charge density or Bader charge — allows for tuning the reduction ability of the active sites, which helps to avoid the deep reduction of adsorbed C₂ intermediates into ethane or ethanol (*J. Mater. Chem. A*, 2024, 12, 31925).

Regarding suggestions for constructing other effective photocatalysts to produce products containing unsaturated bonds, we think multi-site regulation has emerged as a key trend, as single active sites often struggle to break scaling relationships, thereby limiting the formation of multi-carbon products. In contrast, multi-site systems — composed of either single or heteroatomic elements — offer enhanced capabilities for modulating C-C coupling process and stabilizing reaction intermediates, making them more promising for the efficient production of C=C products such as ethylene (*Nat. Catal.* 2022, 5, 564; *J. Am. Chem. Soc.* 2025, 147, 15654). Moreover, the formation of C=C products such as ethylene requires the participation of a substantial amount of H. Therefore, regulating the water dissociation process to ensure sufficient H supply is also a viable strategy. For instance, incorporating non-metallic atoms, like S, Br, F, can promote H generation, thereby enhancing ethylene production (*Nat. Catal.* 2020, 3, 478).

Comment 3: *By the way, the synergistic effects of different metal centers in selectively achieving C=C double bond products or other types of bonding products need to be discussed within the revised manuscript.*

Response: Thanks for the Reviewer's suggestion. We **have cited** the above literatures and **added** corresponding discussion about the synergistic effects of different metal centers in

selectively achieving different types of bonding products in the revised manuscript on page 2 as follows:

“Concurrently, the multi-metal sites on these catalysts exert precise control over the electronic structure of the active site and the adsorption configuration of intermediates during CO₂ reduction, therefore regulating the selectivity to different types of bonding products.^{10, 11} For example, Chen et al. fabricated a new 2D ultrathin bimetallic oxyhalide Pb_{0.6}Bi_{1.4}O₂Cl_{1.4} and achieved a performance for CO₂ photoreduction to methanol.¹² They also utilized the Ag-Cu Lewis acid-base dual sites on the surface of Ag₂Cu₂O₃ nanowires to realize photoconversion of CO₂ to methane.¹³ Besides, multi-metal sites can also break the linear-scaling relationship and promote the generation of multi-carbon (C₂₊) products, like the designed Ni-V sites for ethane production.^{14, 15, 16,}

Comment 4: *What are the space groups and unit cells of the as-synthesized four different composition materials? Why have their high-resolution image been indexed to different crystallographic planes while preserving the same interplanar angle?*

Response: Thanks for Reviewer’s concerns. The space group of Cu₂ZnSnS₄, Cu₂CoSnS₄ and Cu₂MnSnS₄ is I-42m (121) and space group of Cu₂NiSnS₄ is F-43m (216). We **have added** this space group information in the revised manuscript and SI. The experimental crystal lattice parameters according to the XRD results are as follows: a = b = 5.5136 Å, c = 10.826 Å (α = β = γ = 90°) for Cu₂MnSnS₄ (PDF#51-0757); a = b = 5.402 Å, c = 10.805 Å (α = β = γ = 90°) for Cu₂CoSnS₄ (PDF#26-0513); a = b = 5.427 Å, c = 10.848 Å (α = β = γ = 90°) for Cu₂ZnSnS₄ (PDF#26-0575); a = b = c = 5.425 Å (α = β = γ = 90°) for Cu₂NiSnS₄ (PDF#26-0552).

Regarding the different crystallographic planes, we can see that in Fig. 2, the exposed facet of Cu₂ZnSnS₄, Cu₂CoSnS₄ and Cu₂MnSnS₄ can be inferred along [112] direction because interplanar distances match well with the d₋₁₁₀ and d₁₀₋₂ spacings with the corresponding dihedral angle of 60°. But for Cu₂NiSnS₄, due to its different space group, the exposed facet can be inferred along [111] direction because interplanar distances match well with the d₁₁₀ and d₁₀₁ spacings with the corresponding dihedral angle of 60°. We have described this difference in the legend of Fig. 2 in the original manuscript as follows:

“TEM, HRTEM and EDS mapping images for (a) CMTS-S_v nanosheets, (b) CCTS-S_v nanosheet, (c) CNTS-S_v nanosheets and (d) CZTS-S_v nanosheets, in which the exposed facet can be inferred along [112] direction ([111] direction for CNTS-S_v nanosheets) because interplanar distances match well with the d_{-110} and d_{10-2} spacings (d_{110} and d_{101} spacings for CNTS-S_v nanosheets), and the corresponding dihedral angle of 60° agrees well with the calculated angle between the (-110) and (10-2) planes ((110) and (101) planes for CNTS-S_v nanosheets).”

Comment 5: *By the way, are the observed different nanosheet orientations directly related to the calculation model?*

Response: Yes, the observed different nanosheets orientations are directly related to the calculation model. The calculated exposed facet with the lowest surface formation energy for Cu₂ZnSnS₄, Cu₂CoSnS₄ and Cu₂MnSnS₄ is [112] direction and [111] for Cu₂NiSnS₄, consistent with the experimental results from TEM images in Fig. 2. We’ve already explained the relationship of theoretical and experimental results in Supplementary Figure 25.

Comment 6: *How do you consider the atomicity within the different crystal structural models used in calculation compared to the experimental ones? From the structure models shown in the manuscript, both M_i and M_j have well-defined positions; is that consistent with the experimental data? Furthermore, I suggest using different colors to represent M_i and M_j; currently, both are represented by grey, which makes it challenging for readers to distinguish them from the figures.*

Response: Thanks for Reviewer’s concerns and suggestions. The crystal structural models we used in calculations before optimization have the same crystal lattice parameters with the experimental samples obtained from XRD results. And all the bulk models are from the standard crystal databases (ICSD). Since the Cu₂M_iM_jS₄ (M_i=Mn, Co, Ni, Zn; M_j=Sn) is the standard compounds, not like doping or atom replacement, the M_i and M_j do have well-defined positions, which is consistent with the experimental data.

Regarding the colors of the M_i and M_j metal atoms, we **have changed** them to distinctly different colors to distinguish them in the revised manuscript.

Comment 7: *When writing the redundant CO₂ on both sides of Equations 1 and 2? It may be challenging for readers unfamiliar with this area to understand these concepts. Is it also beneficial to include the metal centers that absorb these molecules within the equations? Similar problems also happen for other intermediates.*

Response: Thanks for Reviewer's concerns and suggestions. To clarify the equations for readers, we **have revised** the Equations 1 and 2, and also included the metal centers that adsorb these molecules within the equations.

Comment 8: *The author claimed that sulfur atoms act as an active site for hydrogen adsorption; does any in-situ spectroscopy data support this claim rather than calculation?*

Response: Thanks for the Reviewer's concerns and suggestions. To be honest, we currently only have the calculated results to support the claim that sulfur atoms act as an active site for hydrogen adsorption. To the best of our knowledge, there appears to be no in situ technology that can directly prove hydrogen adsorption on specific atoms.

Point-to-point Response to Reviewer #4

Overall comments: *This manuscript proposed the Bader-Regulate-Performance principle — apposite Bader charge can provide a moderate energy barrier for intermediate adsorption and C-C coupling simultaneously, thus promoting the performance for ethylene generation from CO₂ photoreduction, specifically focusing on the Cu₂MiMjS₄ (Mi=Mn, Fe, Co, Ni, Zn; Mj=Ge, Sn) nanosheets. The Cu₂ZnSnS₄, with the appropriate Bader charge, achieves a record ethylene yield under visible light irradiation. Although **the theoretical expectation is interesting**, several limitations should be considered.*

Overall response: We greatly appreciate the Reviewer's positive comments and kind suggestions.

Comment 1: *In the mechanistic study of the C-C coupling pathway (including the three typical pathways: CO-CO, CO-CHO, and CO-COH), the authors did not provide activation energy barrier calculations for the direct coupling of the CO dimer (CO-CO). The current analysis is based solely on the energy barrier for the formation of the COCOH intermediate via CO and*

*COH, making it difficult to accurately assess the C-C coupling mechanism. It is recommended to supplement the bond formation energy barrier of *CO-*CO coupling and compare it with the hydrogenation-protonation process of *CO-*CO (as these two steps are competitive), to more cautiously determine the most probable coupling pathway and subsequent selection criteria.*

Response: Thanks for the Reviewer's concerns and suggestions. We actually considered all three typical pathways for C-C coupling, including CO-CO, CO-CHO and CO-COH. But we found that C-C coupling through CO-CO is not possible in our system theoretically. As shown in Fig. N4, the built models of coupled CO-CO* intermediate on the Cu pair site will dissociate into two separately adsorbed CO* intermediates after the structure optimization. Regarding the CO-CHO pathway in Fig. N5, we found that the energy barrier of the transition states for this C-C coupling process is 1.52 eV, which is significantly higher than that through the CO-COH pathway (0.91 eV). Therefore, we only included the pathway (CO-COH) with the lowest energy in our original manuscript. To make it clearer and more comprehensive to understand the C-C coupling mechanism, we **have added** all the possible pathways for C-C coupling process in our revised SI as Supplementary Figure 15 and 16.

Fig. N4 | C-C coupling process through CO-CO pathway. (a) Side view and (b) top view of the structure before optimization. (c) Side view and (d) top view of the structure after optimization.

Fig. N5 | C-C coupling process through CO-CHO pathway. (a)-(h) The TS models. (i) The calculated TS energy plots. The TS are calculated by the CI-NEB method, in which the energy barrier is computed to 1.52 eV.

Comment 2: *In the manuscript, the hydrogenation of CO_2 is identified as the rate-determining step (RDS) based on the thermodynamic energy barrier, which is inappropriate. Moreover, the free energy barrier for this process is 1.52 eV, indicating that this step is thermodynamically challenging. However, experimental results show that the CZTS-Sv catalyst exhibits excellent catalytic performance for CO_2 reduction to ethylene, which contradicts the theoretical prediction. It is recommended to verify this discrepancy.*

Response: Thanks for the Reviewer's concerns. There are numerous literatures in which the rate-determining step (RDS) is identified based on the thermodynamic energy barrier [*J. Am.*

Chem. Soc. 2024, 146, 29028; *Nat. Commun.* 2022, 13, 2964; *Nat. Catal.* 2023, 6, 987; *Angew. Chem. Int. Ed.* 2023, 62, e202216613]. Normally, the Gibbs free energy profile is calculated for the specific catalyst and reaction, and the step exhibiting the largest free energy difference is designated as the RDS [*Nat. Synth.* 2025, 4, 53; *Nat. Catal.* 2024, 7, 1000; *Nat. Catal.* 2022, 5, 564]. In our view, this is a proper and justified approach.

Regarding the second question, for most types of CO₂ reduction reactions, energy barriers exist for converting CO₂ into various products [*Nat. Catal.* 2023, 6, 987; *Nat. Synth.* 2025, 4, 53]. To overcome these barriers, external energy input is required. In electrocatalysis, thermocatalysis, and photocatalysis, the driving forces are electricity, heat, and light energy, respectively [*Nat. Catal.* 2023, 6, 987; *Nat. Synth.* 2025, 4, 53; *Angew. Chem. Int. Ed.* 2023, 62, e202302253]. In our case, even though there is a free energy barrier for the process of CO₂ hydrogenation, **driving under the simulated solar irradiation**, the CZTS-Sv catalyst exhibits excellent catalytic performance for CO₂ reduction to ethylene, which does not contradict with the theoretical predictions.

Comment 3: *In lines 312-315 on page 10 of the main text, the authors directly compare the transition state energy barrier for H₂O dissociation (1.38 eV) with the free energy barrier for CO₂ hydrogenation to form the *COOH intermediate (1.52 eV). This approach presents an inconsistency in the evaluation criteria. In fact, the energy barrier comparison for competing reaction pathways should follow one of the following principles: (i) using transition state energy barriers consistently (for kinetically controlled systems), or (ii) using free energy barriers consistently (for thermodynamically controlled systems). It is recommended to recalculate the transition state energy barrier for the CO₂ → *COOH pathway or to use free energy changes uniformly for a cross-comparison to accurately assess the impact of the hydrolysis reaction on CO₂ reduction.*

Response: According to the Reviewer's advice, we **have calculated** the transition state energy barrier for the CO₂ → *COOH pathway as shown in Fig. N6 and **added** the corresponding description in the revised manuscript on page 14-15.

Fig. N6 | TS of $\text{CO}_2 \rightarrow \text{COOH}^*$ processes on the surface of the CNTS-S_v slab. (a)-(h) The TS models. (i) The calculated TS energy plots. The TS are calculated by the CI-NEB method, in which the energy barrier is computed to 1.695 eV.

Comment 4: *In the theoretical screening section of Figure 1c, the authors did not clearly explain the performance advantage criteria for CZTS-S_v within the orange region (e.g., adsorption energy, coupling energy barrier, and other key indicators). Although the manuscript mentions that catalysts in this region exhibit superior properties, the lack of well-defined screening benchmarks weakens the persuasiveness of the theoretical guidance. Furthermore, the experimental section characterizes and tests five catalysts within the screening range, exposing the limitations of the theoretical predictions.*

Response: Thanks for the Reviewer's concerns. In our system, the Bader charge on the unsaturated Cu sites can regulate CO^* adsorption and the C-C coupling processes, enabling the catalyst to provide an appropriate CO^* adsorption energy that increases the local CO concentration while avoiding an excessively high C-C coupling barrier, thereby facilitating the formation of multi-carbon products. Therefore, the Bader charge on the unsaturated Cu sites

can be used as the screening benchmarks to design the catalyst. Based on the above results, we proposed the Bader-Regulate-Performance (BRP) principle in our original manuscript: “*a moderate Bader charge could effectively balance CO* adsorption and C-C coupling processes and optimize them to suitable levels (neither too strong nor too weak), facilitating CO₂ reduction to produce C₂₊ products.*” To clearly explain the performance advantage criteria for CZTS-S_v within the orange region, we **have added** the corresponding expression in the revised manuscript on page 5 as follows: “*The ideal photocatalysts with Bader charge around 0.42 on unsaturated Cu sites in the orange circle area of Fig. 1c, like CZTS slabs, could provide moderate E_{ad}(CO*) and E_{C-C} simultaneously, possessing a higher potential for C₂₊ generation from CO₂ reduction according to the BRP principle.*”

To verify the theoretical predictions, the corresponding catalysts were synthesized and experimentally evaluated for their performance. We **have added** the corresponding explanation to strengthen validation for the theoretical results in the revised manuscript on page 9 as follows: “*Experimental results reveal that CZTS and CMTS nanosheets yield higher C₂H₄ production, while CCTS shows the lowest activity, consistent with theoretical predictions (Fig. 1c). This is attributed to the moderate CO* adsorption energies and C-C coupling barriers of CZTS and CMTS, which favor C₂₊ product formation. In contrast, the strong CO* binding on CCTS could increase the CO* coverage but leads to a higher C-C coupling energy barrier, limiting its C₂₊ activity.*”

To be honest, our current computational and predictive models are currently limited to the Cu₂M_iM_jS₄ (M_i = Mn, Fe, Co, Ni, Zn; M_j = Ge, Sn) system, which imposes certain limitations. We intend to further explore and improve our system in future studies.

Responses to the Reviewers' comments to the manuscript: NCOMMS-25-17898A. We would like to thank all the reviewers for the insightful comments and suggestions, and for their time in helping us to improve this manuscript.

Point-to-point Response to Reviewer #1

Overall comments: *This revised manuscript has addressed all the points raised by reviewers and is now recommended for publication as it is.*

Overall response: We greatly appreciate the reviewer for agreeing to recommend our article for publication.

Point-to-point Response to Reviewer #2

Overall comments: *The author addressed all my comments, I would recommend it publish, thanks.*

Overall response: We thank the Reviewer for agreeing to recommend our article for publication.

Point-to-point Response to Reviewer #3

Overall comments: *My concern has been well addressed within the revised manuscript. It can be accepted for publication. Congratulations to the authors!*

Overall response: We thank the Reviewer for agreeing to recommend our article for publication.

Point-to-point Response to Reviewer #4

Overall comments: The authors have addressed the reviewers' concerns well, and the manuscript has also been improved much. Thus, the paper can be accepted for publication as it is.

Overall response: We thank the Reviewer for agreeing to recommend our article for publication.